# Histological and Molecular Evaluation of Liver Biopsies: A Practical and Updated Review

**DOI:** 10.3390/ijms26167729

**Published:** 2025-08-10

**Authors:** Joon Hyuk Choi

**Affiliations:** Department of Pathology, Yeungnam University College of Medicine, 170 Hyeonchung-ro, Nam-gu, Daegu 42415, Republic of Korea; joonhyukchoi@ynu.ac.kr

**Keywords:** liver, biopsy, pathology, immunohistochemistry, diagnosis

## Abstract

Liver biopsy remains an indispensable diagnostic modality in contemporary hepatology because most classification systems and pathogenetic concepts are grounded in morphology. The diagnostic yield of a biopsy hinges on specimen adequacy and meticulous tissue processing; however, interpretation often challenges even experienced pathologists. This narrative review summarizes practical aspects of histological and molecular assessment for both clinicians and pathologists. Key topics include specimen handling, selection of ancillary stains, recognition of pivotal patterns of hepatic injury, and a systematic approach to differential diagnosis. Mastery of both histological and molecular principles is essential for accurate diagnosis, appropriate therapy, and reliable prognostication.

## 1. Introduction

The liver encounters a broad spectrum of metabolic, toxic, infectious, circulatory, autoimmune, and neoplastic insults [1]. Histopathology therefore remains central to patient care—confirming diagnoses, directing additional investigations, gauging treatment response, and informing prognosis [2]. The evidence guiding biopsy interpretation has accumulated over six decades, beginning with Menghini’s landmark description of percutaneous needle biopsy in 1958 [3,4,5].

Although advances in serology and imaging have reduced the frequency of biopsy in selected scenarios (e.g., acute viral hepatitis or overt biliary obstruction), exclusive reliance on clinical data leads to misclassification in 15–50% of cases [6,7,8]. Non-invasive tools that estimate fibrosis and necroinflammatory activity continue to evolve; nevertheless, none can yet supplant morphologic assessment [9]. Careful microscopic examination of a representative core remains the gold standard for definitive diagnosis.

Numerous clinical scenarios warrant liver biopsy (Table 1) [10,11,12]. When clinical and radiologic data are inconclusive, tissue sampling can resolve diagnostic uncertainty in both children and adults. Biopsy is especially useful for confirming neoplasia, elucidating unexplained jaundice, and investigating fever of unknown origin. Within precision-medicine frameworks, biopsy specimens also support genetic and molecular profiling of tumors, particularly hepatocellular carcinoma (HCC), thereby guiding targeted therapy [13,14].

Despite these benefits, histopathologic assessment is not infallible. The liver displays a restricted spectrum of injury patterns; therefore, many inflammatory changes are nonspecific and may preclude definitive interpretation [15]. Morphologic overlap between benign and malignant lesions further complicates assessment. In addition, sampling error, interpretive variability, procedural risk, and the growing availability of non-invasive tests collectively limit the routine use of biopsy.

Accurate tissue diagnosis underpins appropriate therapy and prognostication. This review provides a comprehensive and practical framework for evaluating liver biopsies, aligned with real-world diagnostic workflows. It begins with an overview of normal histology and biopsy techniques, then covers ancillary methods such as immunohistochemistry (IHC) and molecular testing, which are increasingly incorporated into routine diagnostics. The later sections focus on common patterns of hepatic injury and present diagnostic strategies applicable to everyday clinical practice.

## 2. Normal Histology of the Liver

Familiarity with normal hepatic architecture is a prerequisite for accurate biopsy interpretation [16,17]. The hepatic lobule, the traditional structural unit, is hexagonal with a central vein at its core and portal tracts at the vertices (Figure 1). By contrast, the hepatic acinus represents the functional unit, defined by vascular supply and metabolic gradients. Acinar zones 1 (periportal) through 3 (pericentral) are assigned according to distance from the portal triad and decreasing oxygen tension.

Hepatocytes form one-cell-thick plates radiating from portal tracts to central veins, separated by sinusoids conveying mixed portal and arterial blood. Sinusoids are lined by fenestrated endothelial cells; beneath them, the space of Disse contains hepatocyte microvilli and vitamin A-storing stellate cells that orchestrate fibrosis. Opposing hepatocytic grooves create bile canaliculi (1–2 µm in diameter), which empty into the canals of Hering before reaching ductules and interlobular bile ducts within portal tracts (Figure 2). The canal of Hering harbors stem/progenitor cells (“oval cells” in rodents) that can participate in regeneration after severe injury [18,19].

Age-related histological variations must also be recognized. Pediatric livers commonly display two-cell-thick hepatic plates, glycogenated nuclei, scant lipofuscin, occasional extramedullary hematopoiesis, and heightened iron and copper stores. Conversely, older livers show greater anisocytosis and anisonucleosis, abundant lipofuscin, and increased portal fibrosis or hyalinization [20]. Portal tracts normally contain few lymphocytes, macrophages, and mast cells but virtually no neutrophils or plasma cells; inflammatory cells, however, accumulate with age [21]. Finally, the subcapsular 2 mm zone may contain fibrotic portals with bridging bands, potentially mimicking cirrhosis in superficial samples [22].

## 3. Types of Liver Biopsy Specimens

Pathologists assess liver biopsy specimens acquired by several techniques, each with unique diagnostic strengths and weaknesses. Technique selection depends on the clinical scenario and indication, and every method fulfills a specific role in diagnosis and management (Table 2).

Percutaneous needle biopsy is the most widely used technique because of its simplicity, safety, and diagnostic yield. The core is small—typically 10–30 mm long and 1.2–2 mm wide, representing approximately 1/50,000 of the entire liver mass—so diagnostic adequacy may be limited and sampling error increased. Cutting needles (e.g., Tru-Cut, 16–18 gauge) are now preferred to the traditional suction-type Menghini needle because they provide better-preserved, less fragmented cores [23,24]. Blind percutaneous biopsy remains useful for diffuse liver disease but is insensitive to focal or patchy lesions, including cirrhosis and tumors (Figure 3). Conversely, image-guided biopsies performed with ultrasonography or computed tomography (CT) offer higher diagnostic accuracy for localized hepatic lesions [25,26,27].

Transjugular biopsy accesses the liver via the internal jugular vein and is indispensable when percutaneous biopsy is contraindicated, particularly in patients with coagulopathy [28]. Endoscopic ultrasound (EUS)-guided biopsy is a minimally invasive approach in which tissue is obtained with a needle passed through an echoendoscope. It is especially useful in patients already undergoing EUS for other indications, such as pancreatic assessment, or when percutaneous or transjugular routes are unsuitable.

Fine-needle aspiration (FNA) cytology, especially when performed under radiographic guidance, enables precise sampling of focal lesions and yields material suitable for cytologic and histological evaluation. Although occasionally applied to non-neoplastic disease, its principal value lies in tumor diagnosis—benign or malignant, primary or metastatic—where diagnostic accuracy reportedly spans roughly 80% to >95% [29,30,31].

Wedge biopsies are valuable for lesions missed by percutaneous sampling or identified intraoperatively. These samples provide greater tissue volume and targeted sampling; however, subcapsular sampling may limit accuracy. Specimens should be at least 10–15 mm deep to ensure representative parenchyma.

## 4. Handling of Liver Biopsy Specimens

Accurate handling and processing of liver biopsy specimens are essential for reliable morphological assessment. Core biopsies are prone to drying and must be placed immediately in 10% neutral-buffered formalin. To reduce sampling error, specimen length should exceed 1.5 cm [32]. Processing begins with a detailed gross description that notes color, length, diameter, and fragmentation. The entire specimen should be submitted for histology, wrapped in lens paper or enclosed in a biopsy bag to prevent loss. For wedge specimens, sectioning at 0.2 cm intervals perpendicular to the capsule before fixation is recommended to ensure uniform fixation and representative sampling [33].

Normal liver parenchyma is uniformly brown-tan and is intersected by narrow depressions over portal tracts. Deviations in gross color or texture of a needle core can indicate clinically relevant pathology (Figure 4). Fragmented cores frequently accompany cirrhosis, whereas firm, white cores usually signify neoplastic tissue. Yellow coloration suggests marked steatosis; green suggests cholestasis; brown suggests iron overload; and dark brown-black pigmentation may occur in metastatic melanoma. In infants with jaundice, biopsy tissue warrants additional processing. Snap-freezing is required for downstream molecular studies, and a separate fragment should be fixed in glutaraldehyde for electron microscopy to investigate inherited metabolic defects.

Standard formalin fixation followed by paraffin embedding is sufficient for most liver needle biopsies. For optimal histological assessment, step sections should be cut at several levels throughout the block [33]. Although complete serial sectioning is seldom necessary, it is invaluable when small focal lesions—granulomas, metastatic deposits, or parasites—are suspected. Selected levels are stained with hematoxylin and eosin (H&E) and supplemented with special stains as indicated. Residual unstained slides should be archived for ancillary tests such as IHC or molecular assays.

Liver biopsy specimens now underpin precision hepatology and oncologic decision-making because therapeutic choices increasingly rely on molecular data. Consequently, meticulous preanalytical handling is mandatory: routine processing variables can degrade nucleic acids or proteins, skewing results or rendering samples unusable. Evidence-based recommendations have therefore been issued to standardize preanalytical procedures for tissue and blood specimens, thereby enhancing the reliability and clinical utility of molecular testing [34].

## 5. Routine and Special Stains for Liver Biopsy Specimens

In most laboratories, two H&E-stained levels, an iron stain, and a trichrome stain constitute the core panel for medical liver biopsies [35]. Some centers substitute Sirius red for trichrome because it is more sensitive to fibrosis. At minimum, every biopsy should receive H&E and a fibrosis stain.

Additional stains should be selected according to the clinical question to ensure recognition of relevant histological features [36]. Many centers routinely add Perls’ iron stain to evaluate overload and periodic acid–Schiff with diastase digestion (PAS-D) to screen for α1-antitrypsin deficiency. Some institutions also include reticulin and PAS stains. Reticulin delineates the hepatic scaffold and facilitates the diagnosis of nodular regenerative hyperplasia (NRH), whereas PAS highlights cytoplasmic glycogen and accentuates lobular architecture by negative contrast.

Copper histochemistry, most commonly the rhodanine method, assists in diagnosing Wilson’s disease and chronic cholestatic disorders such as primary biliary cholangitis (PBC), primary sclerosing cholangitis (PSC), prolonged bile duct obstruction, and ductopenia. Oil Red O on frozen sections detects microvesicular steatosis, particularly in acute fatty liver of pregnancy or Reye syndrome. Table 3 lists the special stains routinely used in liver pathology.

## 6. Immunohistochemistry

IHC is integral to interpreting liver biopsy specimens and establishing disease-specific diagnoses. The advent of highly specific monoclonal antibodies, together with increasingly sensitive staining platforms, permits reliable identification of numerous antigens in routinely processed sections. Consequently, IHC supports the diagnosis of a wide spectrum of non-neoplastic and neoplastic liver disorders. Table 4 lists the stains most frequently applied in clinical practice.

### 6.1. Non-Neoplastic Liver Diseases

In non-neoplastic settings, IHC is routinely applied to (1) identify biliary epithelium, (2) assess acinar zonation, (3) detect viral antigens, and (4) demonstrate inclusion bodies associated with storage or inherited metabolic disorders.

Cytokeratin (CK) 7 and CK19 immunostaining is widely used to enumerate bile ducts when ductopenia is suspected, as in graft-versus-host disease or chronic allograft rejection. These markers also quantify ductular reactions in biliary tract disease and chronic viral hepatitis. During prolonged cholestasis, periportal hepatocytes may aberrantly express CK7, reflecting metaplastic or reactive change secondary to bile stasis.

Physiologically, the hepatic acinus exhibits oxygen- and Wnt-dependent zonation that can be visualized by glutamine synthetase, which localizes to hepatocytes surrounding the central vein [37,38]. Both non-neoplastic and neoplastic conditions may perturb this pattern; loss, expansion, or irregularity of glutamine synthetase staining provides a useful morphologic clue.

IHC for hepatitis B surface antigen (HBsAg) and core antigen (HBcAg) is routinely performed in biopsies from seropositive patients to document intrahepatic viral protein expression (Figure 5). Immunostaining also assists in diagnosing systemic, non-hepatotropic viral infections, particularly in immunocompromised individuals and transplant recipients, where confirming viral involvement guides management [39]. Staining for α1-antitrypsin and fibrinogen highlights the characteristic intracytoplasmic inclusions of α1-antitrypsin deficiency and fibrinogen storage disease, respectively.

### 6.2. Neoplastic Liver Diseases

In neoplastic pathology, IHC facilitates identification and classification of both primary and metastatic tumors. HCC is supported by positivity for Hep Par-1, arginase-1, α-fetoprotein, polyclonal carcinoembryonic antigen (CEA), CD10, and glypican-3 [40]. In poorly differentiated tumors—particularly in small needle cores—these markers may be weak or absent because of limited sampling.

For intrahepatic cholangiocarcinoma (ICCA), CK7, CK19, monoclonal CEA, and CA19-9 are commonly employed. CK7 and CK19 are expressed in non-neoplastic bile ducts and in neoplastic cholangiocytes; CK19 additionally marks hepatic progenitor cells. CK19-positive HCCs are associated with aggressive behavior and poorer prognosis [41,42].

Glutamine synthetase typically exhibits a distinctive map-like pattern in focal nodular hyperplasia (Figure 6) [43]. Subclassification of hepatocellular adenoma (HCA) relies on several stains, including liver-fatty-acid-binding protein (LFABP), serum amyloid A (SAA), C-reactive protein (CRP), glutamine synthetase, and β-catenin [44,45,46]. *HNF1A*-inactivated HCA shows loss of LFABP (Figure 7), whereas inflammatory HCA demonstrates SAA and CRP positivity.

IHC remains fundamental for evaluating metastatic liver tumors. By applying a panel of antibodies, pathologists determine both the tumor’s lineage (epithelial, mesenchymal, hematolymphoid, or melanocytic) and its probable organ of origin. In adenocarcinomas and other epithelial neoplasms, the first stratification relies on the CK7/CK20 staining profile [47]. Organ-restricted markers are then added—for example, GATA-binding protein 3 (GATA-3) for breast, thyroid transcription factor-1 (TTF-1) for lung, special AT-rich sequence-binding protein 2 (SATB2) for colorectal, prostate-specific antigen (PSA) for prostate, and paired box gene 8 (PAX8) for renal tumors [48,49]. Such sequential testing narrows the differential diagnosis and pinpoints the primary site.

## 7. Electron Microscopy

Transmission electron microscopy (TEM) still has a niche role in liver pathology, particularly in specific diagnostic dilemmas. It is invaluable for inborn errors of metabolism: storage disorders such as type II glycogenosis, Gaucher disease, and Niemann–Pick disease exhibit characteristic ultrastructural signatures [50,51]. TEM can also visualize viral particles—such as paramyxoviruses or hepatitis C virions—when serologic or culture results are equivocal, even in formalin-fixed tissue [52,53]. Drug-induced injury may likewise be demonstrated, exemplified by lamellar lysosomal phospholipid accumulation in amiodarone toxicity [54]. Although routine IHC has largely supplanted TEM, the technique remains useful when immunophenotypic data are inconclusive and tumor histogenesis is uncertain; neuroendocrine neoplasms, for instance, may reveal neurosecretory granules on TEM [55].

## 8. Molecular Analysis

In situ hybridization (ISH) is performed on liver tissue to detect hepatitis A, B, C, and D viruses, cytomegalovirus, and Epstein–Barr virus. ISH for albumin mRNA is highly specific for hepatocytes and hepatocellular neoplasms [56,57]. The polymerase chain reaction (PCR) can be applied to fresh, frozen, or formalin-fixed paraffin-embedded tissue and is a highly sensitive technique for identifying viral, bacterial, parasitic, and genetic diseases [58].

Gene expression array analysis has become a powerful platform for profiling liver disease [59]. Comparative expression data across diverse neoplastic and non-neoplastic conditions allow investigators to link gene signatures to cellular function, histopathologic change, disease susceptibility, and prognosis or therapeutic response. In particular, profiling has clarified the genetic predisposition to HCC in patients with nonalcoholic steatohepatitis (NASH), illuminating mechanisms of hepatocarcinogenesis [60,61,62].

Next-generation sequencing (NGS) refers to a group of advanced DNA sequencing technologies capable of generating massive amounts of data through highly parallelized processes [63]. In liver pathology, NGS has revolutionized diagnostic and research approaches by enabling comprehensive analysis of genetic mutations, epigenetic alterations, and transcriptomic profiles, thereby enhancing the understanding, classification, and targeted treatment of various hepatic diseases [64]. Table 5 summarizes molecular techniques currently used in liver pathology.

## 9. Predominant Injury Pattern

The principal histological patterns of liver injury include hepatitic, necrotic, steatotic, biliary/cholestatic, vascular, depositional/storage, granulomatous, infiltrative/hematologic, and neoplastic changes. Accurate diagnosis depends on pattern recognition, beginning with the capacity to identify isolated lesions—such as steatosis in an otherwise unremarkable biopsy [65]. Pathologists must also recognize each pattern’s boundaries, namely, the point at which specific histological or clinical findings no longer conform to that category.

### 9.1. Hepatitic Pattern

A hepatitic pattern is characterized by lymphocyte-predominant inflammation of hepatic lobules and/or portal tracts. Infiltrates may be focal, patchy, or diffuse. Depending on the etiology, inflammation can predominate within the lobules or the portal tracts.

#### 9.1.1. Acute and Chronic Hepatitis

Distinguishing acute from chronic hepatitis relies mainly on the duration of aminotransferase elevation; persistence for longer than six months defines chronicity. Histology provides additional clues. Moderate-to-severe diffuse lobular inflammation favors an acute process because such widespread injury cannot be sustained over time. In contrast, established fibrosis reliably indicates chronic hepatitis. Although portal-predominant inflammation often accompanies chronic disease, it is not specific.

Acute-on-chronic hepatitis arises in two principal contexts [65]. First, pre-existing liver disease may be complicated by an independent acute insult—for example, autoimmune hepatitis (AIH) infected by herpes simplex virus. Histology then shows chronic changes (e.g., fibrosis or bile duct injury) plus superimposed high-grade acute necrosis. Second, an intrinsic flare of a chronic disorder such as chronic hepatitis B or AIH produces abrupt enzyme elevation with moderate-to-severe lobular necroinflammation.

#### 9.1.2. Lobular Hepatitis

The severity of lobular hepatitis ranges from minimal to severe and is graded with 10× or 20× objectives, depending on the chosen system [66,67]. In the Ishak scheme, inflammation is semi-quantified by counting foci per 10× field [66]. Moderate-to-severe disease often exhibits confluent zone-3 necrosis that may progress to bridging necrosis. Infiltrates are T-cell-rich with scattered plasma cells; additional features include apoptotic hepatocytes, lobular disarray, Kupffer-cell hyperplasia, hepatocellular rosettes, and cholestasis.

#### 9.1.3. Neutrophilic Inflammation in the Lobules

Neutrophil-rich lobular inflammation arises in two scenarios. First, “surgical hepatitis”, most often seen in resection specimens, shows sinusoidal and perivenular neutrophils—predominantly in zone 3—without hepatocyte injury and carries no clinical significance. Second, scattered neutrophilic clusters (microabscesses) may appear within lobules. These are usually incidental, require no additional workup, and often need not be reported. In transplant biopsies, however, numerous microabscesses are frequently associated with cytomegalovirus infection [68,69].

#### 9.1.4. Portal Tract Inflammation

Portal tract inflammation ranges from mild to severe and is dominated by lymphocytes with variable plasma cells, eosinophils, and histiocytes. Lymphoid aggregates, and occasionally follicles, accompany moderate or greater inflammation. Marked portal plasma cells occur in AIH, drug-induced liver injury (DILI), and some viral hepatitis. Plasma cells are typically IgG-dominant in AIH, whereas PBC contains relatively more immunoglobulin M-positive plasma cells [70,71].

#### 9.1.5. Interface Activity

Interface activity (formerly piecemeal necrosis) denotes inflammation of periportal hepatocytes directly abutting portal tract collagen (Figure 8). The process may be focal or diffuse, and severity varies between tracts; grading therefore reflects the circumferential extent of involvement. Interface activity is a hallmark of the hepatitic pattern yet occurs in multiple acute and chronic disorders, including viral hepatitis, AIH, and DILI. Its presence signifies hepatitic injury but lacks etiologic specificity.

#### 9.1.6. Grading Hepatitis Activity

The severity of hepatitis is gauged by semi-quantitatively scoring its three principal histological features—lobular inflammation, portal inflammation, and interface activity. The Knodell system was the first widely adopted scoring method for chronic hepatitis; it was later refined into the Ishak system to allow a more detailed assessment of necroinflammatory activity and fibrosis stage [66,72]. The Batts–Ludwig and METAVIR systems were subsequently introduced to simplify and standardize the evaluation process, and the METAVIR system clearly separates bridging from non-bridging fibrosis [68,73,74].

#### 9.1.7. Resolving Hepatitis

The resolving hepatitis pattern is usually seen after a mild, self-limited episode of acute hepatitis caused by viral infection or DILI. Biopsy reveals subtle findings, notably small clusters of pigmented macrophages within lobules and occasionally in portal tracts, reflecting prior hepatocellular injury. Portal and lobular inflammation is minimal, and fibrosis is usually absent unless the event arises in acute-on-chronic liver disease.

### 9.2. Necrotic Pattern

Hepatic necrosis manifests in several patterns: spotty, confluent, bridging, zonal, panacinar, submassive, and massive. Spotty necrosis entails isolated or small clusters of dead hepatocytes and is common in both acute and chronic hepatitis (Figure 9). Confluent necrosis usually centers on zone 3 and affects larger clusters of hepatocyte death (≥3 cells); this pattern signals severe lobular injury and can progress to bridging necrosis. Bridging necrosis is a form of confluent necrosis that links distinct lobular regions, typically forming portal-to-portal, portal-to-central, or central-to-central bridges (Figure 10). Zonal necrosis mainly affects zone 3 but can extend to zones 1 or 2. Panacinar necrosis involves multiple adjacent acini. Massive necrosis denotes necrosis exceeding 75% of the liver parenchyma, whereas submassive necrosis affects 25–75% of the parenchyma and may represent a post-necrotic state with regenerative nodules amid collapsed tissue (Figure 11).

### 9.3. Steatotic Pattern

Hepatic steatosis is divided histologically into two primary patterns, macrovesicular and microvesicular. Macrovesicular steatosis, the most common form, is characterized by hepatocytes containing one large lipid droplet that pushes the nucleus to the cell periphery (Figure 12). Cells with multiple or smaller droplets remain categorized as macrovesicular. More than 90% of cases accompany metabolic syndrome or chronic alcohol use, and serum aspartate aminotransferase (AST) and alanine aminotransferase (ALT) levels are typically mildly elevated.

Microvesicular steatosis is marked by hepatocytes packed with numerous small lipid droplets, producing a finely vacuolated cytoplasm while the nucleus remains central (Figure 13). The fat is diffusely distributed, frequently accentuated in zone 3. Microvesicular steatosis is linked to mitochondrial injury and arises in settings such as DILI, toxic exposures, acute fatty liver of pregnancy in adults, and fatty-acid oxidation or urea-cycle disorders in children.

### 9.4. Cytoplasmic Changes and Inclusions in Hepatocytes

#### 9.4.1. Glycogen Accumulation

Histologically, glycogen accumulation presents as diffusely swollen hepatocytes with clear, rarefied cytoplasm. In infants and young children, it usually indicates a glycogen storage disease (Figure 14), whereas in older patients, it most often reflects glycogenic hepatopathy secondary to diabetes mellitus. PAS staining accentuates the stored glycogen and facilitates diagnosis.

#### 9.4.2. Ballooned Hepatocytes

Ballooned hepatocytes are enlarged with pale, flocculent cytoplasm and may harbor Mallory–Denk bodies—ubiquitinated cytoskeletal aggregates (Figure 15). They typify steatohepatitis but also occur in cholestasis or severe acute hepatitis of varied causes. Immunohistochemical markers (p62, ubiquitin, CK8/CK18) accentuate ballooned cells but are unnecessary for routine diagnosis. Ballooning likely reflects cytoskeletal injury and disturbed proteostasis; its mechanism remains elusive [75,76].

#### 9.4.3. Giant-Cell Transformation

Giant-cell transformation, defined by multinucleation and marked enlargement, is more common in children than in adults (Figure 16). It frequently accompanies cholestatic liver disease but can also arise in idiopathic adult giant-cell hepatitis, chronic hepatitis C, AIH, viral infections, DILI, hematologic disorders, and genetic syndromes.

#### 9.4.4. Hepatocyte Inclusions

Multiple cytoplasmic inclusions have diagnostic value. Eosinophilic globules in α1-antitrypsin deficiency—an autosomal recessive disorder caused by *SERPINA1* mutations—are misfolded protein aggregates that stain magenta with PAS-D and localize to zone 1 hepatocytes. Megamitochondria appear in fatty liver disease and certain inherited metabolic disorders. Ground-glass and pseudoground-glass inclusions may be seen in chronic hepatitis B and drug-induced injury, respectively [77].

### 9.5. Biliary/Cholestatic Patterns

#### 9.5.1. Biliary Obstructive Pattern

Bile duct obstruction may arise from stones, strictures, or external compression by pancreatic or other peribiliary masses. Histological findings correlate more with the duration and severity of obstruction than with its cause. Acute obstruction shows bile ductular proliferation, mixed portal inflammation with lymphocytes, neutrophils, and eosinophils, and occasional portal edema. In longstanding obstruction, ductular proliferation and portal inflammation persist but are usually less pronounced.

#### 9.5.2. Ductular Reaction

The ductular reaction is a stereotyped response to cholestatic or hepatocellular injury—bile duct obstruction, PBC, PSC, DILI, and others [78]. The key features are (1) portal edema, (2) proliferation of ductular structures at the limiting plate, and (3) focal neutrophilic infiltrates, often adjacent to ductules (Figure 17). The proliferating ductules are thought to arise from periportal stem/progenitor cells within the canals of Hering [79,80].

#### 9.5.3. Ductopenia

Ductopenia (bile duct loss) most often accompanies chronic cholestatic disorders such as PBC and PSC but may also follow DILI, paraneoplastic syndromes, or chronic allograft rejection [79,80,81,82]. When no cause is identified, the process is termed idiopathic ductopenia or vanishing bile duct syndrome [83]. Lobular cholestasis may be absent. Histologically, ductopenia denotes loss of bile ducts in ≥50% of portal tracts; therefore, assessment requires at least ten well-preserved portal tracts. Unpaired hepatic arteries and absent ducts in medium-to-large portal tracts are additional clues. CK7 immunostaining underscores duct loss and highlights zone-1 intermediate hepatocytes.

#### 9.5.4. Bland Lobular Cholestasis

Bland lobular cholestasis is defined by accumulation of bile pigment in hepatocytes or bile canaliculi without significant portal/lobular inflammation, steatosis, or hepatocellular injury (Figure 18). Portal tracts are unremarkable, lacking biliary obstruction or duct loss. The pattern is most frequently related to DILI, although sepsis and other rare etiologies should also be considered. The exact compartment of pigment deposition—hepatocytic versus canalicular—has no diagnostic consequence.

#### 9.5.5. Ascending Cholangitis

Ascending cholangitis represents a bacterial infection of the biliary tree, usually superimposed on structural abnormalities or immunosuppression. Diagnosis is predominantly clinical, and biopsy is seldom required. When performed, histology shows dilated portal ducts with attenuated epithelium and intraluminal neutrophils, often accompanied by obstructive changes.

### 9.6. Vascular (Circulatory) Disease Patterns

Hepatic vascular disorders arise from reduced inflow, disturbed intrahepatic flow, or obstructed outflow. Cirrhosis is the most common cause of impaired intrahepatic perfusion. Histologically, four principal patterns are recognized: portal vein disease, sinusoidal/centrilobular disease, vascular outflow obstruction, and peliosis hepatis.

#### 9.6.1. Idiopathic Noncirrhotic Portal Hypertension

Idiopathic noncirrhotic portal hypertension (INPH) is diagnosed when clinical or radiologic evidence of portal hypertension occurs in the absence of underlying parenchymal liver disease [84]. Affected portal veins exhibit dilatation, herniation, atrophy, fibrosis, or thrombosis—changes collectively termed obliterative portal venopathy [85]. Additional findings may include isolated intralobular bile ducts, NRH, and portal tract crowding resulting from hepatocyte atrophy [86].

#### 9.6.2. Veno-Occlusive Disease

Damage to sinusoids and central veins produces veno-occlusive disease, also called sinusoidal obstructive syndrome (SOS). Histology demonstrates patchy sinusoidal dilatation with or without thrombosis or fibrous obliteration of central veins. SOS develops mainly after allogeneic (less often autologous) hematopoietic stem cell transplantation or in patients treated with specific chemotherapeutic agents [87,88,89].

#### 9.6.3. Vascular Outflow Obstruction Disease

Vascular outflow obstruction results from blockage of central veins or the inferior vena cava or from chronic right-sided heart failure. Histologically, the hallmark is zone-3 sinusoidal dilatation, with or without congestion. This finding is not specific and also accompanies portal vein disease, hematologic disorders (e.g., sickle cell anemia), systemic inflammatory diseases (e.g., sarcoidosis, rheumatoid arthritis), paraneoplastic syndromes, and oral contraceptive use [90]. Patchy bile ductular proliferation may mimic biliary obstruction.

#### 9.6.4. Peliosis Hepatis

Peliosis hepatis consists of cyst-like, blood-filled cavities scattered throughout the lobule without zonal predilection [91]. Lesions may be focal or diffuse and initially lack an endothelial lining, which may develop over time. Cavities vary in size, are usually non-communicating, and can contain blood, serum, or loose collagen in chronic stages. Although generally unrelated to vascular inflow or outflow disorders, peliosis hepatis occasionally accompanies INPH [92,93]. Rupture, though uncommon, can cause intraperitoneal hemorrhage [94].

### 9.7. Depositional/Storage Disorders

Chronic liver diseases frequently produce hepatic iron and copper accumulation, which serves as a histological marker of longstanding injury.

#### 9.7.1. Iron

Hepcidin, a liver-derived peptide, is the principal regulator of systemic iron homeostasis. In noncirrhotic livers, marked iron overload is usually caused by hereditary hemochromatosis. Conversely, chronic liver diseases reduce hepcidin synthesis or activity, leading to parenchymal iron deposition, especially when fibrosis is advanced [95]. Secondary iron overload from exogenous sources (e.g., repeated transfusion) predominantly involves Kupffer cells. Iron is readily demonstrated with Perls’ Prussian blue stain (Figure 19).

#### 9.7.2. Copper

Copper is excreted primarily in bile. In Wilson’s disease, hepatic copper distribution is heterogeneous throughout the disease course; some regenerating cirrhotic nodules may contain no copper. This heterogeneity can produce false-negative rhodamine stains or quantitative assays; therefore, sampling at least two liver cores is recommended to mitigate diagnostic error (Figure 20) [96]. In cholestatic disorders (e.g., PBC, PSC), impaired bile flow causes periportal copper accumulation, a useful histological clue.

#### 9.7.3. Amyloid

Amyloidosis denotes extracellular deposition of misfolded protein fibrils that disrupt tissue function. The liver is frequently involved in systemic amyloidosis. Light-chain (AL) amyloid usually arises from plasma-cell dyscrasias (e.g., multiple myeloma), whereas amyloid A (AA) follows chronic inflammation. AL typically deposits in sinusoids, whereas AA more often involves blood vessels. Congo red with apple-green birefringence remains the diagnostic gold standard. IHC can distinguish AA, AL, and transthyretin amyloid, and mass spectrometry-based proteomics definitively classifies all types [97,98].

### 9.8. Granuloma

A granuloma comprises epithelioid histiocytes (activated macrophages) encircled by lymphocytes and often multinucleated giant cells. In the liver, granulomas reside in portal tracts or lobules. In Western cohorts, leading causes include PBC, idiopathic disease, sarcoidosis, infection, and DILI. Subtypes include lipogranuloma (steatotic liver disease), fibrin-ring (Q fever), non-caseating (sarcoidosis, PBC, DILI), caseating (tuberculosis), necrotic (bacterial, fungal, parasitic infection; DILI), and foreign-body granuloma (injection sites or surgery).

### 9.9. Fibrosis

Fibrosis drives morbidity and mortality in chronic liver diseases, including viral hepatitis, DILI, steatotic, biliary, and vascular disorders. It typically starts in portal tracts but may arise in lobules or central veins, especially in steatotic and outflow disorders. Persistent injury activates hepatic stellate cells, which transdifferentiate into collagen-producing myofibroblasts. Progression depends on etiology and accelerates once bridging fibrosis appears. Although non-invasive methods are gaining acceptance, histological assessment with trichrome or Sirius red stains remains the gold standard. Reports may use formal staging systems (e.g., METAVIR, Ishak, Batts–Ludwig) or descriptive architecture (pericellular, portal, bridging, cirrhosis) with modifiers such as focal, diffuse, mild, or moderate.

#### 9.9.1. Pericellular Fibrosis

Pericellular fibrosis (perisinusoidal fibrosis) usually arises in zone 3 and is most often linked to steatotic liver disease. It features thin, irregular collagen strands best seen with trichrome stain (Figure 21). The NASH Clinical Research Network scores pericellular fibrosis as mild when detectable only by trichrome staining and moderate when also apparent on H&E staining [67,99].

#### 9.9.2. Central Vein Fibrosis

Central vein (perivenular) fibrosis is most frequent in alcohol-related liver disease and venous outflow disorders. Because it often coexists with pericellular fibrosis, it is rarely scored separately in histological staging systems.

#### 9.9.3. Portal/Periportal Fibrosis

Portal fibrosis typically presents as enlarged, irregular portal tracts with collagenous extensions that entrap hepatocytes. Although the terms portal and periportal fibrosis are often used interchangeably, portal fibrosis denotes collagen within the portal tract, whereas periportal fibrosis emphasizes deposition around the tract. Common pitfalls include overcalling fibrosis in chronic hepatitis and mistaking inflammatory or ductular portal expansion for fibrosis. When findings are equivocal, a lack of collagen on trichrome stain supports a designation of no fibrosis.

#### 9.9.4. Bridging Fibrosis

Bridging fibrosis is defined by fibrotic bands connecting portal tracts and/or central veins, forming a mesh-like network within the liver parenchyma [100,101]. Although central-to-central and central-to-portal bridging were once considered prognostically distinct, such differences are rarely discernible in biopsy specimens and are now grouped together. Key diagnostic pitfalls include tangential sectioning of portal tracts, bridging necrosis, and severe lobular fibrosis, all of which can mimic true bridging fibrosis. Accurate diagnosis depends on recognizing dense, linear fibrotic bands characteristic of true bridging fibrosis and distinguishing them from irregular, inflamed, or collapsed tissue that may resemble fibrosis.

#### 9.9.5. Cirrhosis

Cirrhosis denotes diffuse replacement of normal parenchyma by fibrotic bands and regenerative nodules across the liver (Figure 22). Band thickness depends on disease activity and chronicity, ranging from thin to thick. Traditional classifications based on nodule size (micronodular vs. macronodular) are no longer clinically relevant. Key diagnostic challenges include (1) advanced fibrosis lacking distinct nodules, particularly in active alcoholic steatohepatitis—when fibrosis is diffuse, the pattern is considered cirrhosis even without discrete nodules—and (2) specimen fragmentation, now more common with small-gauge-needle biopsies. Fragmentation should not be used as a diagnostic criterion for cirrhosis. Staging should therefore rely on observed architecture, supplemented by a comment outlining limitations imposed by fragmentation [102,103,104]. Cirrhosis is now viewed as a potentially reversible stage in the dynamic, bidirectional course of chronic liver disease rather than an irreversible endpoint [105].

Several histological scoring systems have been developed to evaluate the extent of necroinflammatory activity (grading) and fibrosis (staging) in chronic hepatitis [106,107,108,109,110,111,112]. Table 6 shows a comparison of these systems. Table 7 summarizes the principal histological patterns of liver injury and their representative causes.

## 10. Molecular Pathology in Liver Biopsies

Molecular pathology has become increasingly integral to the evaluation of liver biopsy specimens. Notably, substantial progress has been achieved in the molecular characterization of hepatic tumors, particularly in their diagnosis, their classification, and the identification of potential targets for therapy [45,113,114]. In parallel, an expanding spectrum of genetically defined cholestatic liver diseases is now widely recognized [115]. Moreover, genetic mutations may contribute to drug-induced cholestasis because the liver plays a vital role in the metabolism of drugs and xenobiotics. Table 8 provides a summary of representative molecular features associated with both non-neoplastic and neoplastic liver lesions.

## 11. Diagnostic Approach to Liver Pathology

### 11.1. Clinical and Laboratory Information

Accurate interpretation of a liver biopsy requires a thorough understanding of the clinical context. The specific indication for the biopsy must be clearly defined. Pathologists should integrate all relevant clinical and laboratory data into their diagnostic assessment.

### 11.2. Pathological Diagnostic Approach

Pathologists evaluate morphological abnormalities in ways that directly inform diagnosis and management. A biopsy may not always be diagnostic and can sometimes add little clinical value. Nevertheless, when properly obtained and processed, it remains a cornerstone of hepatic assessment.

Accurate histological interpretation requires a systematic approach [116]. Evaluation begins at low magnification, surveying overall architecture and the spatial relationships among portal tracts, central veins, and hepatic plates. Stains such as reticulin or trichrome highlight fibrosis or parenchymal collapse, revealing subtle changes suggestive of cirrhosis or NRH.

After assessing global architecture, each hepatic compartment should be examined systematically. A practical sequence starts with the portal tracts, evaluating the hepatic artery, portal vein, bile ducts, and associated inflammatory infiltrates. Lobules are subsequently assessed for steatosis, cholestasis, inflammation, and glycogen accumulation, whereas hepatocytes are inspected for inclusions or pigments. Sinusoids are examined for dilatation, endothelial changes, or extracellular deposits including amyloid or fibrosis. Finally, central veins are scrutinized for inflammation or other lesions. Table 9 outlines this systematic method of liver biopsy examination.

Because the liver exhibits a restricted set of morphological responses to injury, histology must be integrated with clinical, biochemical, immunological, and imaging data. Although some pathologists prefer to review slides before receiving clinical data to minimize bias, reports must remain comprehensive and clinically oriented. Structured reports, ideally using standardized checklists or diagnostic summaries, improve clarity, consistency, and clinical utility.

Biopsies obtained for unexplained enzyme elevation or ascites may appear nearly normal histologically [117]. In certain cases, injury produces no microscopic correlate. In others, damage is mild, patchy, and missed because of sampling error. Pathologists must therefore examine such specimens meticulously to avoid missing subtle clues. Despite exhaustive evaluation, the etiology of enzyme elevation remains unexplained in roughly 25% of ostensibly normal biopsies [118].

## 12. Future Perspectives

Emerging molecular and digital technologies are redefining the role of liver biopsy. Fresh tissue, formalin-fixed paraffin-embedded tissue samples, touch imprints, and archival blocks all support genomic analyses [119]. Technologies such as next-generation sequencing and transcriptomics allow detailed characterization of mutational landscapes and expression signatures in liver disease [120,121]. These tools are especially valuable in oncology, where somatic mutation profiling guides targeted therapy for liver cancers [122]. They also refine subclassification and prognostication in NAFLD and AIH [123,124].

Digital pathology is reshaping diagnostic practice through whole-slide imaging, artificial intelligence-based quantification of fibrosis and steatosis, and algorithmic evaluation of tissue architecture [125,126]. Collectively, these tools enhance reproducibility, enable remote consultation, and underpin the integration of histology with multi-omic datasets.

Non-invasive diagnostic tests such as elastography (e.g., FibroScan, MR elastography), serum biomarkers (e.g., fibrosis-4, aspartate aminotransferase-to-platelet ratio index), and standard imaging techniques (e.g., ultrasound, conventional or contrast-enhanced MRI) are increasingly used to assess liver fibrosis and disease severity; however, they cannot provide detailed architectural, cellular, or etiological information [127]. Therefore, liver biopsy remains indispensable for diagnosing both non-neoplastic and neoplastic liver diseases [128]. Even minute tissue fragments can provide critical information that guides therapy, refines prognostication, and identifies actionable biomarkers or molecular targets. In the era of precision medicine, histopathologic findings directly influence a wide range of therapeutic decisions. For instance, histological activity and fibrosis scores play a key role in initiating or discontinuing treatment for conditions such as chronic hepatitis B, AIH, and metabolic dysfunction-associated steatotic liver disease (MASLD), particularly with emerging antifibrotic therapies [129,130]. Furthermore, liver biopsy specimens are increasingly incorporated into clinical trial protocols for pharmacodynamic biomarker analysis [131,132]. Although some debate its current role, liver biopsy remains a cornerstone of patient care, and liver pathology continues to evolve in alignment with modern clinical demands [133].

## 13. Conclusions

Despite significant advancements in non-invasive diagnostics, liver biopsy remains the gold standard for diagnosing a wide range of liver disorders, including chronic hepatitis, drug-induced liver injury (DILI), and neoplastic or post-transplant conditions. Histological assessment complements clinical, laboratory, and imaging data; however, accurate interpretation requires integration with the clinical context and close collaboration between clinicians and pathologists. Advances in genomics, transcriptomics, and digital pathology, as well as the growing use of molecular testing in routine diagnostics, are reshaping the role of liver biopsies. Molecular analysis can aid in disease classification, identify actionable mutations, and reveal inherited or acquired genetic etiologies, particularly in neoplastic and cholestatic liver diseases. As these technologies continue to advance, the diagnostic, prognostic, and therapeutic utility of liver biopsies—and of liver pathology more broadly—will be further enhanced in the era of precision hepatology.

## Figures and Tables

**Figure 1 ijms-26-07729-f001:**
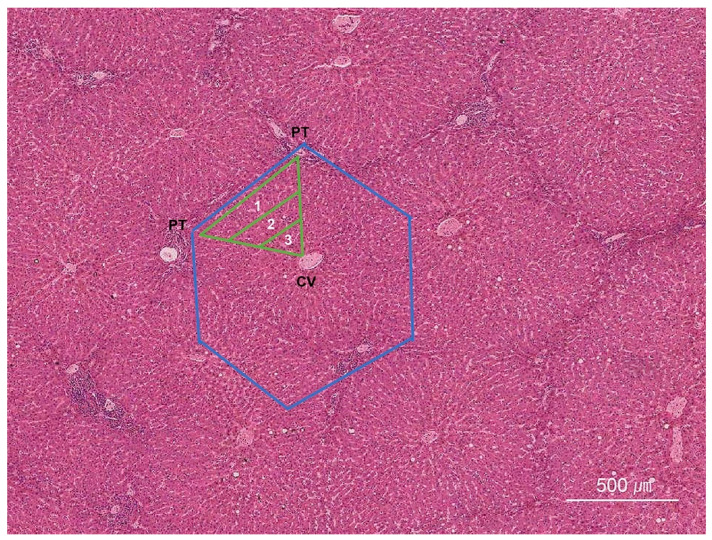
The hepatic lobular and acinar architecture. In the hepatic lobular model, the central vein (CV) is at the center of a lobule, while the portal tracts (PTs) are at the periphery. In contrast, the acinar model—based on the direction of blood flow—divides the hepatic parenchyma into three zones: zone 1 is closest to the blood supply, whereas zone 3 is the most distal. The blue lines indicate lobules, which are divided into centrilobular, midzonal, and periportal zones. The green lines indicate sinusoids, which are divided into zones 1, 2, and 3 (H&E stain, scanning power).

**Figure 2 ijms-26-07729-f002:**
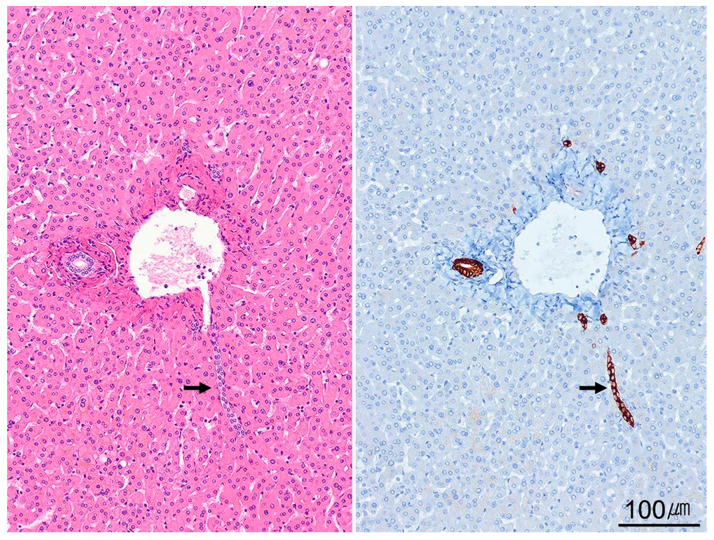
Portal tract and canal of Hering. The portal tract contains a bile duct, a hepatic artery, and a portal vein embedded within dense collagen (**left**). The canal of Hering is a transitional ductule partially lined with hepatocytes and partially with cuboidal cholangiocytes (arrow). Bile flows from the canal of Hering into the intrahepatic bile ductules, which are entirely lined with cholangiocytes. Cytokeratin 19 immunostaining highlights the canal of Hering (arrow) (**right**). The canal of Hering serves as a niche for liver stem/progenitor cells (H&E stain, ×100; cytokeratin 19 immunostain, ×100).

**Figure 3 ijms-26-07729-f003:**
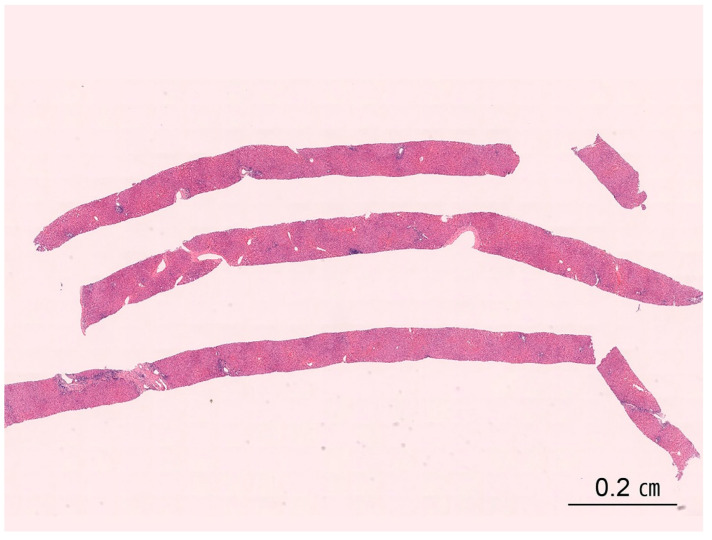
Percutaneous needle biopsy specimen. Three cores of liver tissue are shown. An adequate liver biopsy specimen should be approximately 2 cm in length and include 11 to 15 portal tracts to ensure reliable histopathological evaluation (H&E stain, scanning power).

**Figure 4 ijms-26-07729-f004:**
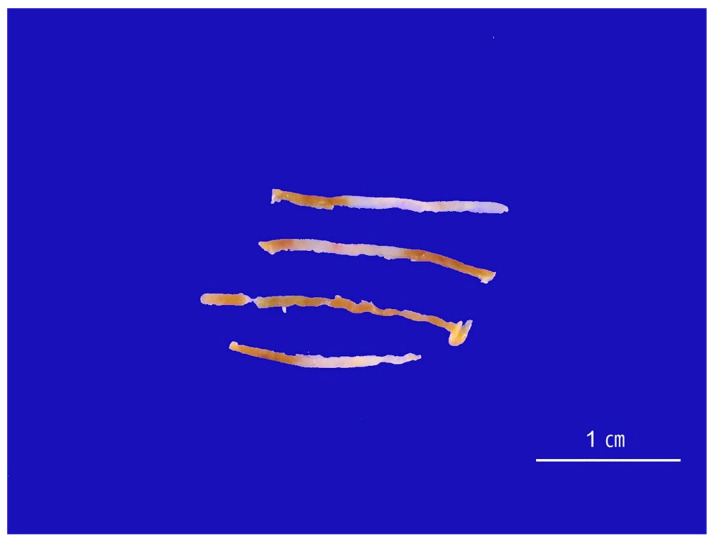
Gross appearance of the needle biopsy specimen. The biopsy cores demonstrate a grayish-white lesion. Histological and clinical correlation confirmed metastatic carcinoma originating from the breast.

**Figure 5 ijms-26-07729-f005:**
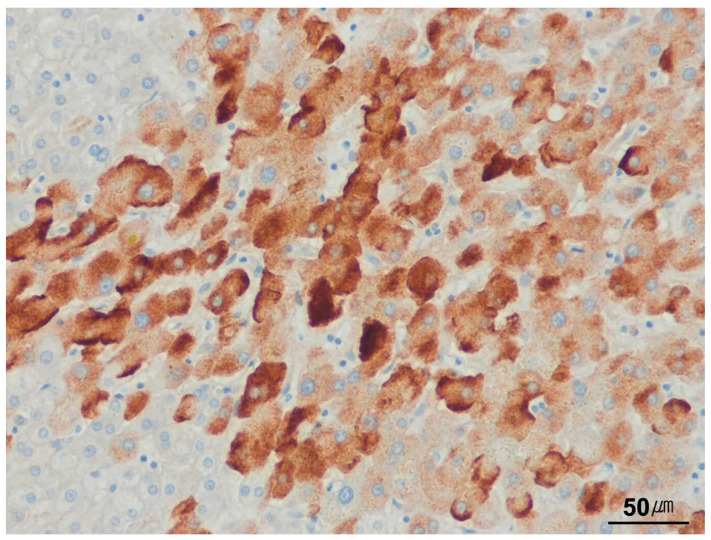
Hepatitis B surface antigen in chronic hepatitis B. Hepatitis B surface antigen (HBsAg) reactivity is seen in the cytoplasm of hepatocytes in a case of chronic hepatitis B (HBsAg immunostain, ×200).

**Figure 6 ijms-26-07729-f006:**
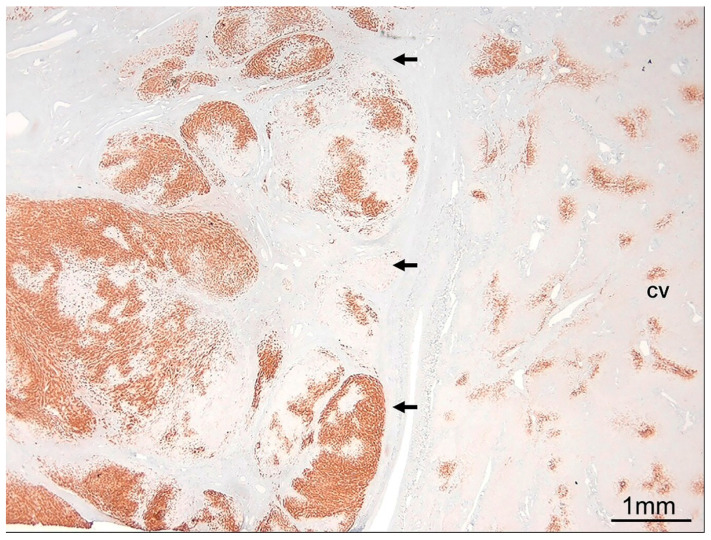
Focal nodular hyperplasia. Immunohistochemical staining for glutamine synthetase demonstrates the characteristic anastomosing, map-like pattern within the lesion (arrows), which is typical of focal nodular hyperplasia. In contrast, the adjacent non-lesional liver parenchyma exhibits physiological zone 3 staining around the central vein (CV) (glutamine synthetase immunostain, ×1).

**Figure 7 ijms-26-07729-f007:**
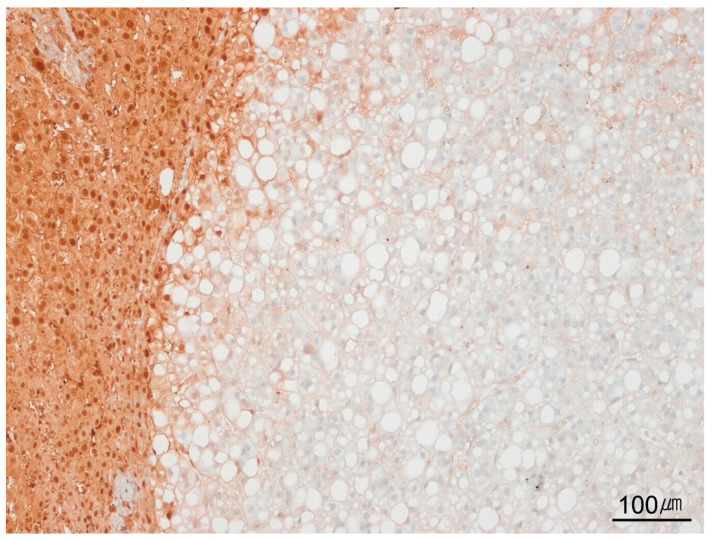
*HNF1A*-inactivated hepatocellular adenoma. Immunohistochemical staining for liver-fatty-acid-binding protein (LFABP) shows complete loss of expression in tumoral hepatocytes (**right**), with preserved expression in adjacent non-tumoral liver tissue (**left**) (LFABP immunostain, ×100).

**Figure 8 ijms-26-07729-f008:**
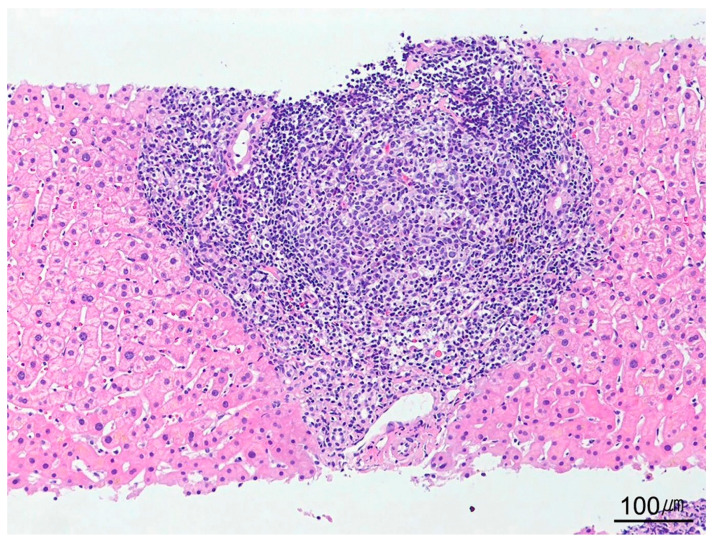
Interface hepatitis. The portal tract shows interface activity involving more than 50% of its circumference, characterized by inflammatory cell infiltration at the limiting plate (H&E stain, ×100).

**Figure 9 ijms-26-07729-f009:**
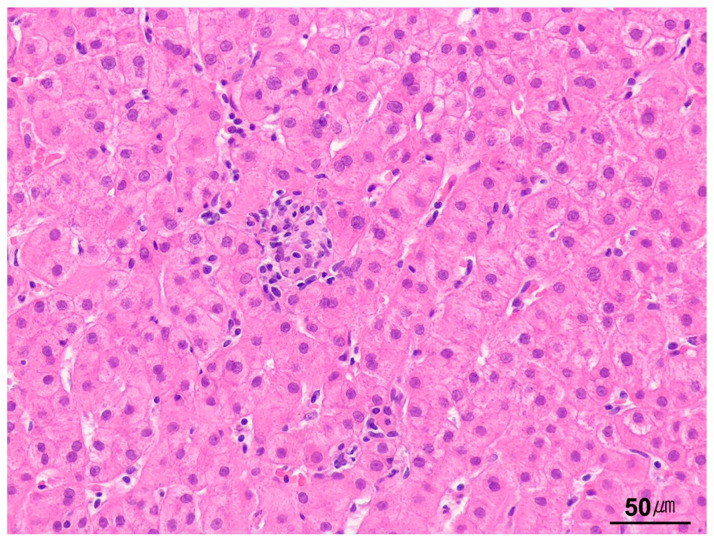
Spotty necrosis. Spotty (focal) necrosis appears as focal hepatocyte dropout accompanied by lymphocytic infiltration (H&E stain, ×200).

**Figure 10 ijms-26-07729-f010:**
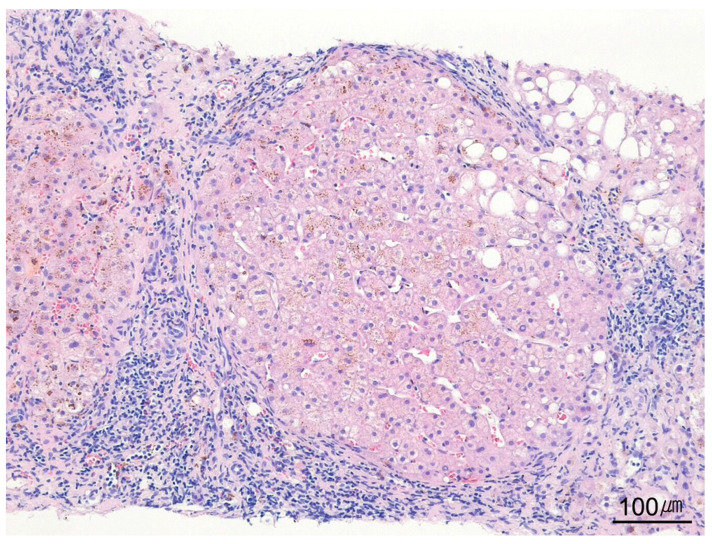
Bridging necrosis. A narrow zone of hepatocyte loss and inflammation extends between adjacent portal tracts in a case of chronic hepatitis (H&E stain, ×100).

**Figure 11 ijms-26-07729-f011:**
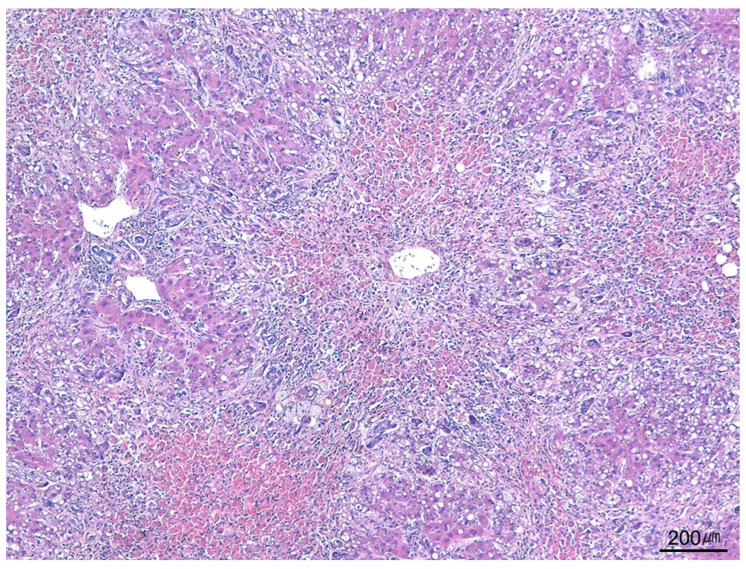
Submassive necrosis. Submassive necrosis predominantly involving zone 3 hepatocytes is observed in a case of acetaminophen toxicity (H&E stain, ×40).

**Figure 12 ijms-26-07729-f012:**
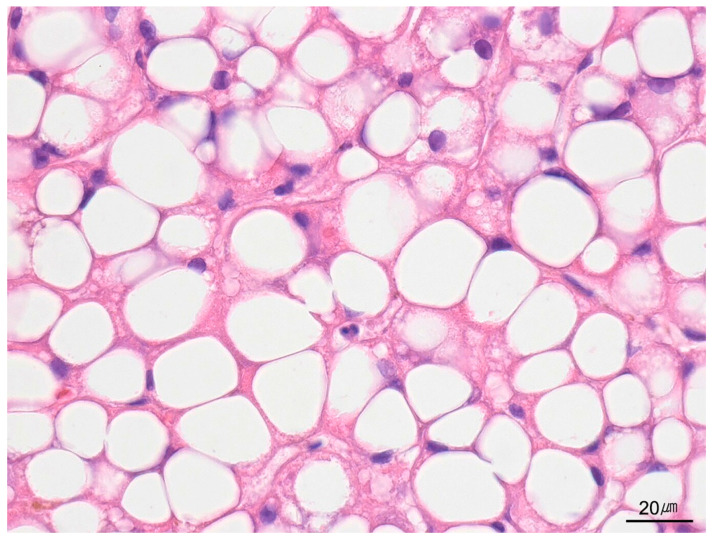
Macrovesicular steatosis. Hepatocytes contain large cytoplasmic fat droplets that displace the nucleus to the periphery, characteristic of macrovesicular steatosis (H&E stain, ×400).

**Figure 13 ijms-26-07729-f013:**
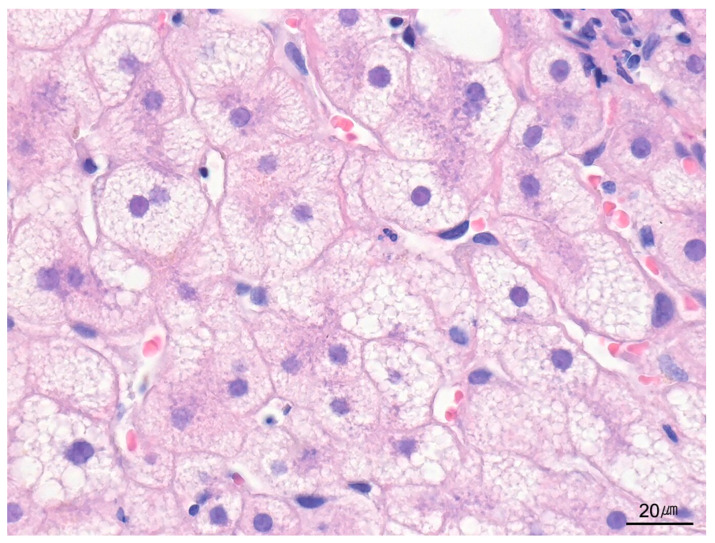
Microvesicular steatosis. Hepatocyte cytoplasm is diffusely involved by numerous small lipid vesicles, consistent with microvesicular steatosis (H&E stain, ×400).

**Figure 14 ijms-26-07729-f014:**
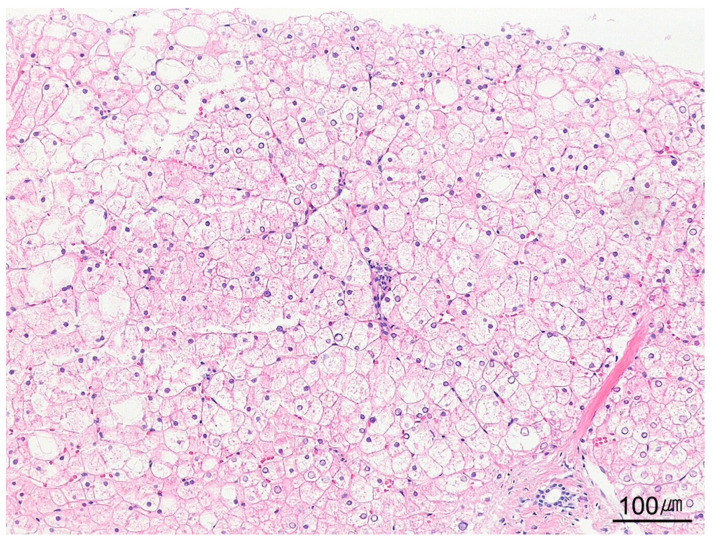
Glycogenic hepatopathy. Hepatocytes appear enlarged and pale due to excessive intracytoplasmic glycogen accumulation in a case of glycogen storage disease (H&E stain, ×100).

**Figure 15 ijms-26-07729-f015:**
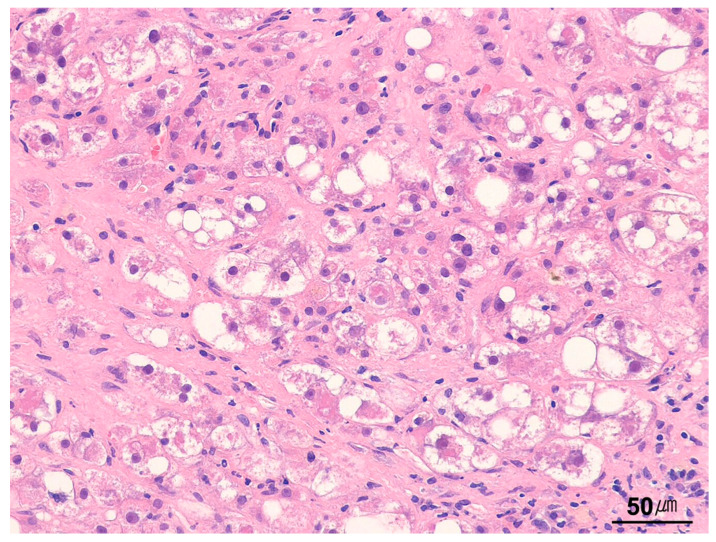
Ballooning degeneration. Ballooned hepatocytes are enlarged with rarefied cytoplasm and show condensation of cytoplasmic proteins forming Mallory–Denk bodies in a case of nonalcoholic steatohepatitis (H&E stain, ×200).

**Figure 16 ijms-26-07729-f016:**
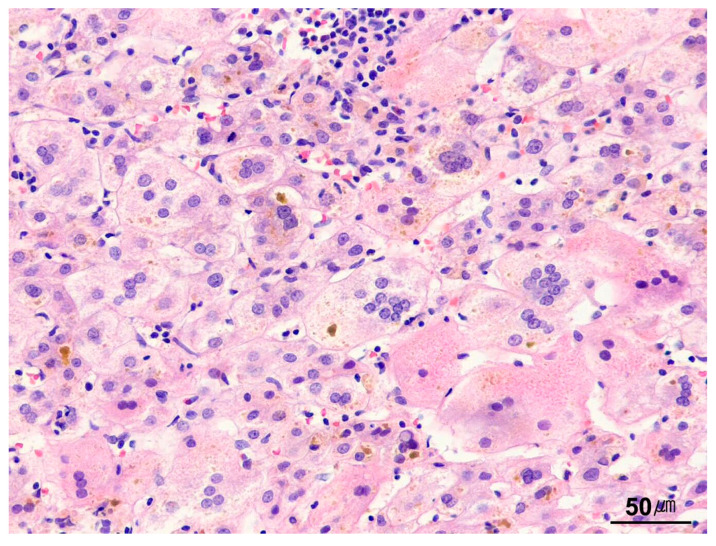
Giant-cell transformation. Multinucleated giant hepatocytes, along with canalicular and hepatocellular cholestasis, are observed in a case of neonatal hepatitis (H&E stain, ×200).

**Figure 17 ijms-26-07729-f017:**
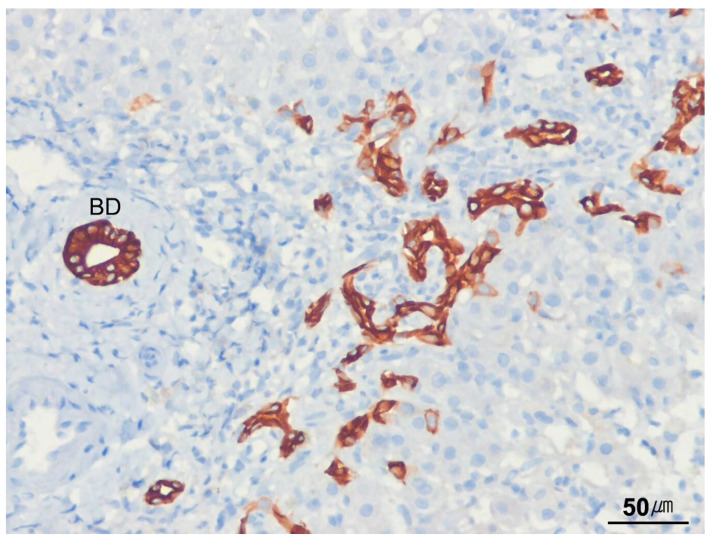
Ductular reaction. Bile ductular proliferation is present. The native bile duct (BD) is identified (cytokeratin 7 immunostain, ×200).

**Figure 18 ijms-26-07729-f018:**
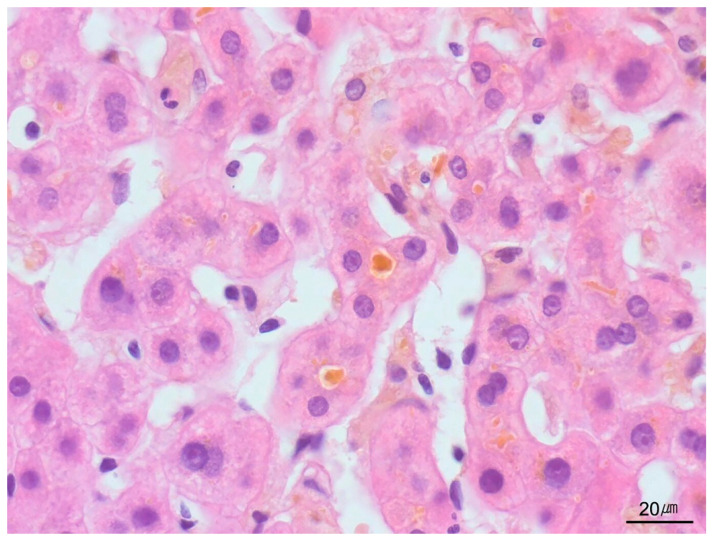
Bland lobular cholestasis. Prominent bile accumulation in the dilated bile canaliculi is seen in a case of drug-induced liver injury. There is no significant associated lobular inflammation or hepatocyte injury (H&E stain, ×400).

**Figure 19 ijms-26-07729-f019:**
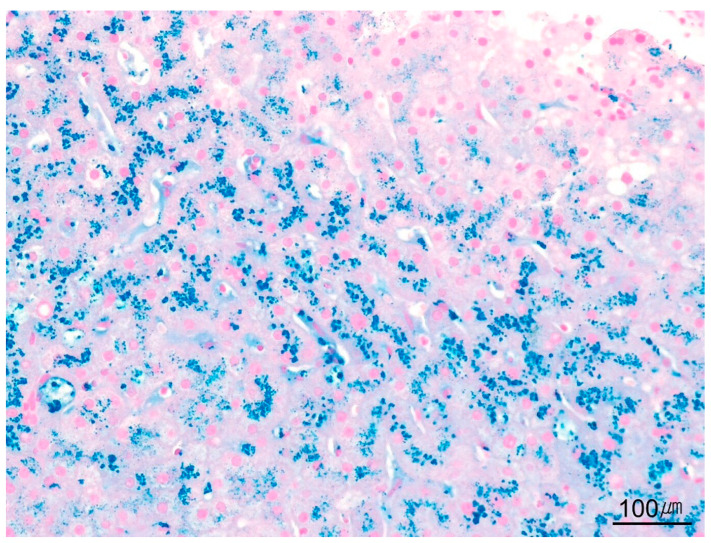
Iron deposition. Prussian blue stain demonstrates iron deposition (Prussian blue stain, ×100).

**Figure 20 ijms-26-07729-f020:**
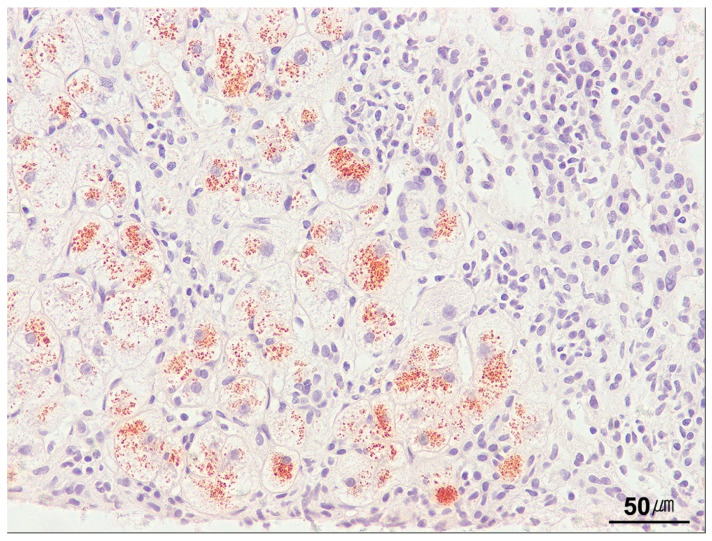
Copper accumulation. Copper accumulation in periportal hepatocytes is observed in a case of Wilson’s disease. Copper appears as a coarse, orange-brown pigment on rhodanine staining (rhodanine stain, ×200).

**Figure 21 ijms-26-07729-f021:**
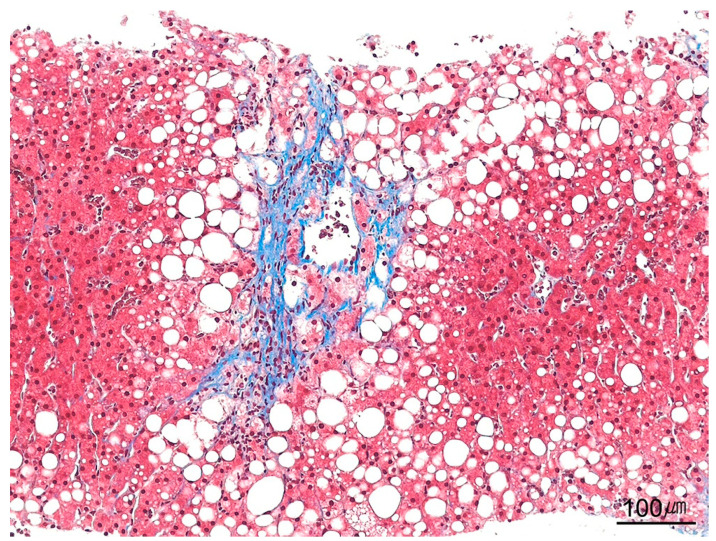
Pericellular fibrosis. Pericellular fibrosis is highlighted by trichrome staining in a case of nonalcoholic steatohepatitis, outlining individual hepatocytes in a chicken-wire pattern (trichrome stain, ×100).

**Figure 22 ijms-26-07729-f022:**
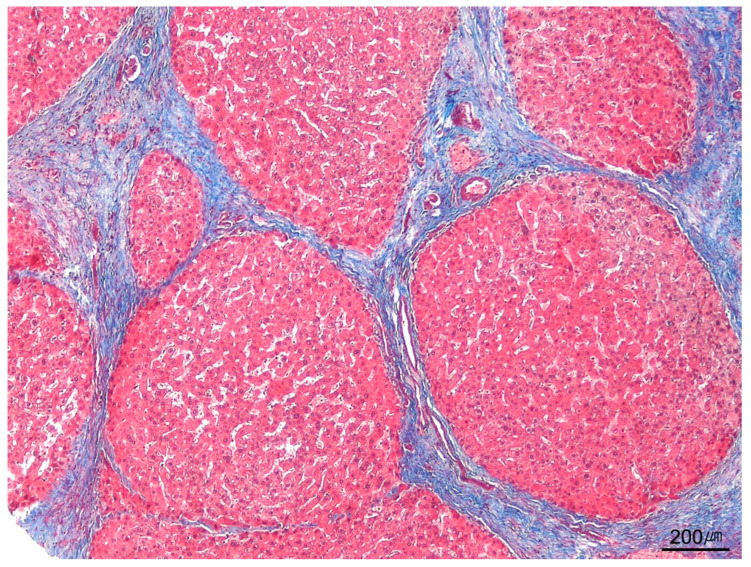
Cirrhosis. Trichrome stain highlights cirrhotic nodules surrounded by broad fibrous septa in a case of alcoholic cirrhosis (trichrome stain, ×40).

**Table 1 ijms-26-07729-t001:** Common indications for liver biopsy.

Category	Indication
Diagnostic evaluation and etiologic determination	Unexplained elevation in liver function tests
	Fever of unknown origin with suspected hepatic involvement
	Diagnostic confirmation and characterization of hepatic neoplasms
	Persistent jaundice of unknown origin
	Suspected inherited metabolic liver disorders
	Acute liver failure of unknown etiology
Disease staging and prognostic assessment	Histological grading and staging of chronic hepatitis (viral, autoimmune, drug-induced)
	Quantification of steatosis and steatohepatitis
	Assessment of ascites or portal hypertension
	Fibrosis staging in candidates for combined organ transplantation (e.g., heart–liver)
Post-transplant and therapeutic monitoring	Unexplained graft dysfunction after solid-organ or hematopoietic stem cell transplantation
	Histological monitoring of treatment response or drug-related hepatic injury

**Table 2 ijms-26-07729-t002:** Liver biopsy techniques.

Category	Technique	Note
Percutaneous	Suction	Widely used; may produce fragmented cores in fibrotic liver (e.g., Menghini, Klatskin, Jamshidi needles)
	Cutting	Yields more intact cores but increases hemorrhage risk (e.g., Vim–Silverman, Tru-Cut needles)
	Spring-loaded	Automated devices delivering uniform core length and rapid acquisition (e.g., ASAP gun)
Image-guided	Thin needle (ultrasound- or CT-guided)	Enables precise targeting of focal lesions; suitable for FNA or core biopsy
Transvenous	Transjugular	Introduced through the internal jugular into the hepatic vein; preferred in coagulopathy; cores may include venous wall
EUS-guided	Transgastric	Reaches lesions inaccessible to the percutaneous approach; particularly valuable for small or deep masses
Laparoscopic or surgical	Laparoscopic	Provides direct visualization and targeted sampling of focal or superficial lesions
	Operative wedge	Performed during laparotomy; yields large tissue volumes but may overrepresent subcapsular parenchyma
Cytologic	FNA	Diagnoses benign versus malignant neoplasms; accuracy is high when it is image-guided

ASAP, automated spring-activated biopsy; CT, computed tomography; EUS, endoscopic ultrasound; FNA, fine-needle aspiration.

**Table 3 ijms-26-07729-t003:** Special stains commonly used in liver pathology.

Special Stain	Diagnostic Application	Staining Color and Components
Collagens and fibrin		
Masson’s trichrome	Detection of type I collagen; assessment of fibrosis stage in chronic hepatitis	Type I collagen: blue; megamitochondria: red; Mallory–Denk bodies: red
Sirius red	Detection of type I/III collagen; staging and fibrosis assessment	Type I/III collagen: red ^(a)^
Gordon and Sweet’s reticulin	Detection of type III collagen; evaluation of lobular architecture, hepatocyte plate thickness, and discrimination of benign versus malignant nodules (e.g., cirrhosis, NRH, HCC)	Reticulin: black; collagen: rose
Phosphotungstic acid hematoxylin	Detection of fibrin in fibrin-ring granuloma (e.g., Q fever, hepatitis A)	Fibrin: blue-purple; collagen: red
Pigments and minerals		
Perls’ Prussian blue	Detection of hemosiderin iron in overload disorders, including hemochromatosis and hemosiderosis	Ferric iron: blue
Hall’s bile	Detection of bilirubin in cholestatic liver diseases and tumors	Bilirubin: green
Rhodanine	Detection of copper in Wilson’s disease and chronic cholestasis	Copper: reddish-orange
Fontana–Masson	Detection of melanin, lipofuscin, and Dubin–Johnson pigment	Melanin, lipofuscin, Dubin–Johnson pigment: black
Glycogen		
PAS	Detection of glycogen, basement membrane, and fungi	Glycogen, basement membrane, fungi: magenta
PAS-D	Highlights intracytoplasmic globules in α1-antitrypsin deficiency and ceroid-laden macrophages	Glycoprotein, basement membrane, α1-antitrypsin globules, ceroid-laden macrophages: magenta; glycogen: cleared
Amyloid		
Congo red	Detection of amyloid in vessels, portal tracts, and perisinusoidal spaces	Amyloid: salmon pink to red (apple-green birefringence under polarized light)
Lipids		
Oil Red O	Detection of lipid ^(b)^	Fat: red; nuclei: blue
Microorganisms		
Ziehl–Neelsen	Detection of acid-fast bacilli	Mycobacterium spp. (e.g., *M. tuberculosis*, *M. leprae*): red
Shikata orcein	Detection of HBsAg	HBsAg: dark brown
Victoria blue	Detection of HBsAg	HBsAg: blue
Grocott methenamine silver	Detection of fungi and selected bacteria ^(c)^	Fungi, selected bacteria: black
Warthin–Starry	Detection of spirochetes	Spirochetes (e.g., *Treponema pallidum*): black

NRH, nodular regenerative hyperplasia; HCC, hepatocellular carcinoma; PAS, periodic acid–Schiff; PAS-D, periodic acid–Schiff with diastase digestion; HBsAg, hepatitis B surface antigen. ^(a)^ Under polarized light, type I collagen exhibits yellow to red birefringence, whereas type III collagen displays green birefringence. ^(b)^ Requires fresh or frozen tissue because fat is dissolved during routine paraffin embedding. ^(c)^ Selected bacteria (e.g., *Nocardia*, *Actinomyces*) may stain.

**Table 4 ijms-26-07729-t004:** Immunohistochemical stains commonly used in non-neoplastic and neoplastic liver lesions.

Immunohistochemical Stain	Diagnostic Application
Identification of bile duct and bile ductules	
Cytokeratin 7 or 19	Highlights bile ducts and ductules; aids assessment of ductopenia and ductular reaction
Identification of liver zonation	
Glutamine synthetase	Marks metabolic zonation, with characteristic peri-terminal venular hepatocyte staining
Identification of viral antigens	
HBsAg and HBcAg	Confirms hepatitis B infection and indicates HBV replication activity ^(a)^
Hepatitis D virus	Detects hepatitis D coinfection or superinfection in HBV-infected patients
Non-hepatotropic viruses	Detects cytomegalovirus, herpes simplex virus, EBV, adenovirus, and others, particularly in immunocompromised patients
Hereditary and storage diseases	
α1-antitrypsin	Highlights eosinophilic intracytoplasmic globules in α1-antitrypsin deficiency
Fibrinogen	Demonstrates hepatocytic fibrinogen accumulation in fibrinogen storage disease
Identification and classification of primary liver tumors	
Hep Par-1	Positive in 70–85% of HCC ^(b)^
Arginase-1	Positive in 45–95% of HCC ^(c)^
α-fetoprotein	Positive in 30–40% of HCC and >90% of hepatoblastomas, particularly fetal and embryonal subtypes
Polyclonal CEA	Positive in 45–80% of HCC ^(d)^
CD10	Positive in 50–75% of HCC ^(d)^
Glypican-3	Positive in 50–80% of HCC ^(e)^
CD34	Shows diffuse sinusoidal staining in HCC
Cytokeratin 7 and 19 ^(f)^	Positive in 70–90% of ICCA
Monoclonal CEA	Positive in ~60% of ICCA
CA19-9	Positive in ~60% of ICCA
Glutamine synthetase	Displays a map-like pattern in focal nodular hyperplasia
LFABP	Loss of expression in H-HCA
Serum amyloid A	Positive in inflammatory HCA
C-reactive protein	Positive in inflammatory HCA
β-catenin	Displays nuclear positivity in B-HCA exon 3
CD31 and ERG	Positive in angiosarcoma
SMA and HMB45	Positive in PEComa (angiomyolipoma)

HBsAg, hepatitis B surface antigen; HBcAg, hepatitis B core antigen; HBV, hepatitis B virus; EBV, Epstein–Barr virus; HCC, hepatocellular carcinoma; ICCA, intrahepatic cholangiocarcinoma; CEA, carcinoembryonic antigen; LFABP, liver-fatty-acid-binding protein; H-HCA, hepatocyte nuclear factor 1α-inactivated hepatocellular adenoma; B-HCA exon 3, β-catenin-activated hepatocellular adenoma with exon 3 mutations; PEComa, perivascular epithelioid cell tumor. ^(a)^ HBcAg positivity indicates active HBV replication. ^(b)^ Hep Par-1 outperforms arginase-1 in well-differentiated HCC. ^(c)^ Arginase-1 is superior in poorly differentiated HCC. ^(d)^ Polyclonal CEA and CD10 produce a bile canalicular pattern. ^(e)^ Glypican-3 positivity helps distinguish HCC from dysplastic or cirrhotic nodules. ^(f)^ Cytokeratin 19 expression in HCC correlates with aggressive behavior and poor prognosis.

**Table 5 ijms-26-07729-t005:** Molecular techniques in liver pathology.

Technique	Diagnostic Applications
In situ hybridization	Detection of hepatotropic and non-hepatotropic viruses (e.g., HBV, HCV, EBV, CMV)
	Detection of albumin mRNA in hepatocellular neoplasms
Polymerase chain reaction	Quantification and genotyping of HBV and HCV
	Detection of infectious agents (e.g., viruses, bacteria, parasites)
	Detection of genetic mutations (e.g., *HFE* in hereditary hemochromatosis, *ATP8B1* in progressive familial intrahepatic cholestasis)
	Characterization of hereditary and metabolic liver disorders
Microarray analysis	Gene expression profiling across liver disease subtypes
	Comparative genomic hybridization for chromosomal aberrations
	Detection of single-nucleotide polymorphisms linked to disease susceptibility and prognosis
Next-generation sequencing	Detection of cholestasis-related genes Identification of mutations in HCC (e.g., *TP53*, *CTNNB1*, *TERT* promoter) Comprehensive genomic profiling of biliary tract cancers for targeted therapy Detection of inherited liver diseases (e.g., Wilson’s disease, α1-antitrypsin deficiency)

HBV, hepatitis B virus; HCV, hepatitis C virus; EBV, Epstein–Barr virus; CMV, cytomegalovirus; HCC, hepatocellular carcinoma.

**Table 6 ijms-26-07729-t006:** Comparison of scoring systems for grading and staging of chronic hepatitis.

System, Year of Publication	Grading (Necroinflammatory Activity)	Staging (Fibrosis)	Note	Reference
Knodell (HAI), 1981	Periportal/bridging necrosis (score 0–10); intralobular degeneration and focal necrosis (score 0–4); portal inflammation (score 0–4)	Score 0–4 (no fibrosis to cirrhosis)	Introduced HAI; fibrosis included in total score (0–22); composite scoring can obscure the distinction between inflammation and fibrosis	[72]
Scheuer, 1991	Portal/periportal necroinflammatory activity (grade 0–4); lobular necroinflammatory activity (grade 0–4)	Grade 0–4 (no fibrosis to cirrhosis)	Simplified semi-quantitative assessment; independently scores necroinflammation and fibrosis	[106]
Desmet, 1994	Piecemeal necrosis and lobular activity (grade: minimal to severe) ^(a)^	Score 0–4 (no fibrosis to cirrhosis)	Chronic hepatitis is diagnosed based on etiology, grade, and stage	[107]
Ishak (modified HAI), 1995	Interface hepatitis (piecemeal necrosis) score 0–4); confluent necrosis (score 0–6); focal lytic necrosis/apoptosis/focal inflammation (score 0–4); portal inflammation (score 0–4); total score 0–18	Score 0–6 (no fibrosis to cirrhosis)	Improves Knodell HAI by separating grade/stage and adding detailed fibrosis scoring; standardized and reproducible histological assessment	[66]
Batts–Ludwig, 1995	Interface hepatitis (piecemeal necrosis) (grade 0–4); lobular inflammation and necrosis (grade 0–4)	Score 0–4 (no fibrosis to cirrhosis)	Provides a simplified, practical framework for grading and staging chronic hepatitis	[108]
METAVIR, 1996	Interface hepatitis (piecemeal necrosis) (score 0–3); lobular necrosis (score 0–2); overall histological activity (A0–A3) ^(b)^	Score F0–F4 (no fibrosis to cirrhosis)	Simplified scoring system; fibrosis evaluation is reliable	[73,74]
Park et al., 1999	Porto-periportal activity (score 0–4); lobular activity (score 0–4) ^(c)^	Score 0–4 (no fibrosis to cirrhosis)	Practical and reproducible; tailored for Korean clinical and pathological practice	[109]
Laennec, 2000	Not graded	Score F0–F4 (no fibrosis to cirrhosis); subdividing cirrhosis (F4) into F4A, F4B, or F4C ^(d)^	Correlated strongly with both the clinical stage of cirrhosis and the grade of portal hypertension	[110,111,112]

HAI, histological activity index; A, activity; F, fibrosis stage. ^(a)^ Knodell HAI is used for grading (e.g., HAI 1–3 = minimal, HAI 4–8 = mild, HAI 9–12 = moderate, and HAI 13–18 = severe). ^(b)^ METAVIR histological activity: A0 = no activity, A1 = mild activity, 2 = moderate activity, and A3 = severe activity. ^(c)^ Porto-periportal activity is graded by assessing interface hepatitis (piecemeal necrosis) and portal inflammation, while lobular activity is graded based on lobular inflammation and necrosis. ^(d)^ Cirrhosis (F4) is subdivided into F4A (mild), F4B (moderate), or F4C (severe) according to the thickness of fibrous septa and the size of regenerative nodules.

**Table 7 ijms-26-07729-t007:** Common histological patterns of liver injury and their representative causes.

Category	Histological Pattern	Representative Causes
Hepatitic pattern	Lobular hepatitis Portal tract inflammation Interface activity (hepatitis)	Viral hepatitis (HBV or HCV) Autoimmune hepatitis Drug-induced liver injury (DILI)
Necrotic pattern	Spotty necrosis Confluent necrosis Zonal necrosis Panacinar necrosis Submassive necrosis Massive necrosis	Acute viral hepatitis DILI Ischemia
Steatotic pattern	Macrovesicular steatosis Microvesicular steatosis	Alcohol-associated liver disease NAFLD/NASH (metabolic syndrome) DILI
Cytoplasmic changes	Glycogen accumulation Ballooned hepatocyte Giant-cell transformation	Glycogen storage disease Steatohepatitis Cholestatic liver diseases
Hepatocyte inclusions	Eosinophilic globules Megamitochondria Ground-glass inclusion Pseudoground-glass inclusion	α1-antitrypsin deficiency Fatty liver disease Chronic hepatitis BDILI
Biliary/cholestatic pattern	Biliary obstructive pattern Ductular reaction Ductopenia Bland lobular cholestasis Ascending cholangitis	Biliary stones or stricture PBC, PSC, DILI Biliary obstruction, PBC, PSC, DILI DILI or sepsis Bacterial infection of the biliary tract
Vascular pattern	Portal vein disease Sinusoidal/central vein disease Vascular outflow disease Peliosis hepatis	Idiopathic noncirrhotic portal hypertension Veno-occlusive disease Budd–Chiari syndrome Right heart failure
Depositional/storage pattern	Iron Copper Amyloid	Primary or secondary hemochromatosis Wilson’s disease Amyloidosis
Granuloma	Lipogranuloma Fibrin-ring granuloma Non-caseating granuloma Caseating granuloma Necrotic granuloma Foreign-body granuloma	Steatotic liver disease Q fever Sarcoidosis, PBC, DILI Tuberculosis Bacterial, fungal, or parasitic infections, DILI Drug injection or prior surgery
Fibrosis pattern	Pericellular fibrosis Central vein fibrosis Portal/periportal fibrosis Bridging fibrosis Cirrhosis	Steatotic liver disease Steatotic liver disease; vascular outflow disorders Chronic liver diseases (viral hepatitis, DILI, steatotic, biliary, or vascular disorders)
Infiltrative/hematologic pattern	Infiltration of sinusoids or portal tracts by atypical cells	Malignant lymphoma Leukemia Histiocytic disorders
Neoplastic pattern	Tumor infiltration with distortion of architecture	Primary liver tumors (HCC, ICCA) Metastatic carcinoma

HBV, hepatitis B virus; HCV, hepatitis C virus; NAFLD, nonalcoholic fatty liver disease; NASH, nonalcoholic steatohepatitis; PBC, primary biliary cholangitis; PSC, primary sclerosing cholangitis; HCC, hepatocellular carcinoma; ICCA, intrahepatic cholangiocarcinoma; DILI, drug-induced liver injury.

**Table 8 ijms-26-07729-t008:** Representative molecular features of non-neoplastic and neoplastic liver lesions.

Liver Lesions	Molecular Features
Cholestatic liver diseases	
Alagille syndrome	*JAG1* and *NOTCH2* mutations
αl-antitrypsin deficiency	*SERPINA1* mutations
Cystic fibrosis	*CFTR* mutations
PFIC-1	*ATP8B1* mutations
PFIC-2	*ABCB11* mutations
PFIC-3	*ABCB4* mutations
PFIC-4	*TJP2* mutations
PFIC-5	*NR1H4* mutations
PFIC-6	*MYO5B* mutations
BRIC	*ATP8B1* mutations (BRIC-1) and *ABCB11* mutations (BRIC-2)
ICP	*ABCB4*, *ABCB11*, *ABCC2*, *ATP8B1*, and *NR1H4* mutations
HCA	
H-HCA	*HNF1A* biallelic inactivating mutations
I-HCA	*IL6ST*, *FRK*, *STAT3*, *GNAS*, and *JAK1* mutations
B-HCA exon 3	*CTNNB1* exon 3 activating mutations
B-HCA exon 7/8	*CTNNB1* exon 7 or 8 activating mutations
BI-HCA	Share features of both B-HCAs and I-HCAs
SH-HCA	Somatic deletions of *INHBE*
HCC	
Steatohepatitic HCC	*TP53* mutations
MM HCC	*TP53* mutations and *FGF19* amplification
Scirrhous HCC	*TSC1/2* mutations
CH HCC	Alternative lengthening of telomeres
FL carcinoma	*DNAJB1::PRKACA* fusion gene
Intrahepatic CCA	
Small-duct type	*BAP1* and *IDH1/2* mutations, *FGFR2* fusions, and *SMAD4*, *BAP1*, *BRAF*, *ARIDA1A*, *KRAS*, *TP53*, and *SMAD4* mutations
Large-duct type	*KRAS*, *TP53*, and *SMAD4* mutations and *MDM2* amplification

PFIC, progressive familial intrahepatic cholestasis; BRIC, benign recurrent intrahepatic cholestasis; ICP, intrahepatic cholestasis of pregnancy; HCA, hepatocellular adenoma; H-HCA, hepatocyte nuclear factor 1α-inactivated HCA; I-HCA, inflammatory HCA; B-HCA exon 3, β-catenin-activated HCA with exon 3 mutations; B-HCA exon 7/8, β-catenin-activated HCA with exon 7/8 mutations; BI-HCA, β-catenin-activated inflammatory HCA; SH-HCA, sonic hedgehog-activated HCA; HCC, hepatocellular carcinoma; MM HCC, macrotrabecular massive hepatocellular carcinoma; CH HCC, chromophobe hepatocellular carcinoma; FL carcinoma, fibrolamellar carcinoma; CCA, cholangiocarcinoma.

**Table 9 ijms-26-07729-t009:** Systematic approach to the histological assessment of liver biopsy specimens.

**Low-power inspection** Specimen adequacy: length, fragmentation, number of portal tracts Overall architecture Distortion Irregularity Parenchymal nodularity Fibrosis: presence and pattern General overview Variegation Focal lesions Anatomic distribution of abnormalities **High-power examination** Lobules Inflammation: type, distribution, severity Cholestasis: hepatocellular, canalicular Fibrosis: location, pattern Hepatocytes Injury and necrosis: type, distribution, extent Cytoplasmic changes: steatosis, glycogen, inclusions, pigments Hepatic plates: hyperplasia, regeneration, atrophy Central veins Inflammation: perivenular, venular wall Occlusion: thrombosis, fibrosis Mural thickening: sclerosis Portal tract in general Size and profile: enlargement, irregularity Inflammation: type, distribution, extent Stromal change: edema, fibrosis Periportal alterations: interface activity, fibrosis Bile ducts Number: loss, paucity of bile ducts Epithelial injury: damage, necrosis, degeneration Inflammation: periductal or intraepithelial Periductal fibrosis: fibrous expansion, concentric onionskin Ductal bile plugs: inspissated bile Bile ductules Proliferation: ductular reaction Inflammation: neutrophils, mononuclear cells Ductular bile plugs: luminal bile accumulation Hepatic arteries and portal veins Caliber abnormalities: reduced diameter, dilatation, duplication Inflammation: vasculitis, perivascular infiltrates Thrombosis: vascular occlusion, recanalization Fibrosis: perivascular, mural

## Data Availability

All data are included in the manuscript.

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
