# Peer review of "Histological and Molecular Evaluation of Liver Biopsies: A Practical and Updated Review"

_ijms, 2025, doi:10.3390/ijms26167729_

Round 1

Reviewer 1 Report

Comments and Suggestions for Authors

In general this is quite nice piece of work. I have rather technical remarks. 

For sure like for me manuscript has misleading title as this are not recent advances but rather state of the art or critical review of the present possibilities or diagnostic capabilities... 

Presentation is nice, organization is clear however I don't underatand order of particular subtitles, moreover how to put in one line histology, molecular testing and then injury pattern; diagnostic approach... 

I would consider other types of figures or alternative organization of tables to make it easy reading for readers. 

Moreover, apart of "liver biopsies" as it stands it title, I would expect in such comprehensive review some data concerning other methods of non-invasive diagnostics... As an alternative to invasive ones...

Probably a good idea would be presentation images in form of panel. Perfect would be having serial sections for IHC, what is very informative, however difficult from technical point of view. 

Author Response

Comment 1: In general this is quite nice piece of work. I have rather technical remarks. For sure like for me manuscript has misleading title as this are not recent advances but rather state of the art or critical review of the present possibilities or diagnostic capabilities.

Response 1: Thank you for your comment regarding the title. I agree that “recent advances” may be misleading, as the manuscript provides a comprehensive overview of current diagnostic approaches. Accordingly, I have revised the title to:

“Histological and Molecular Evaluation of Liver Biopsies: A Practical and Updated Review”

Comment 2: Presentation is nice, organization is clear however I don't understand order of particular subtitles, moreover how to put in one line histology, molecular testing and then injury pattern; diagnostic approach.

Response 2: Thank you for your thoughtful comment and for your positive feedback on the manuscript’s presentation and clarity. Regarding your concern about the order of the sections, I would like to clarify that the structure reflects the logical flow of real-world diagnostic practice, as outlined in the Introduction as follows:

“This review provides a comprehensive and practical framework for evaluating liver biopsies, aligned with real-world diagnostic workflows. It begins with an overview of normal histology and biopsy techniques, then covers ancillary methods such as immunohistochemistry (IHC) and molecular testing, which are increasingly incorporated into routine diagnostics. The later sections focus on common patterns of hepatic injury and present diagnostic strategies applicable to everyday clinical practice.”

Comment 3: I would consider other types of figures or alternative organization of tables to make it easy reading for readers.

Response 3: Thank you for your helpful suggestion. I have carefully reviewed all figures and tables to improve clarity and readability. Where appropriate, I have also considered alternative formats and reorganized certain elements to enhance visual impact. Additionally, adjustments were made to font size, alignment, and layout to improve overall presentation and make the content more accessible to readers.

Comment 4: Moreover, apart of "liver biopsies" as it stands it title, I would expect in such comprehensive review some data concerning other methods of non-invasive diagnostics... As an alternative to invasive ones...

Response 4: Thank you for your insightful comment. While a comprehensive discussion of non-invasive diagnostic methods is beyond the scope of this pathology-centered review, I have acknowledged their importance by adding the following sentence to the Future Perspectives section.

“Noninvasive diagnostic tests such as elastography (e.g., FibroScan, MR elastography), serum biomarkers (e.g., fibrosis-4, aspartate aminotransferase to platelet ratio index), and standard imaging techniques (e.g., ultrasound, conventional or contrast-enhanced MRI) are increasingly used to assess liver fibrosis and disease severity; however, they cannot provide detailed architectural, cellular, or etiological information [120].”

Comment 5: Probably a good idea would be presentation images in form of panel. Perfect would be having serial sections for IHC, what is very informative, however difficult from technical point of view.

Response 5: Thank you for your thoughtful suggestion. I agree that panel images and serial IHC sections would enhance clarity and educational value. However, due to technical limitations and limited tissue availability, they could not be included in this review. I appreciate your recommendation and will consider incorporating such features in future work.

Reviewer 2 Report

Comments and Suggestions for Authors

I have had the opportunity to review manuscript, "Recent Advances in Histological and Molecular Evaluation of Liver Biopsies: An Updated Review," and would like to share my feedback. Overall, I believe this is an excellent work that deserves rapid publication.
 The work provides a deep and detailed analysis of the key aspects of liver biopsy evaluation. This is particularly valuable for trainees and specialists who are learning this subject. The text is written in a very clear and structured manner, making it accessible to a wide range of readers. The quality of the illustrations significantly enhances the reader's understanding and is a major strength of this manuscript.
Recommendations for improvement: While the work is already very strong, I believe it could be further enhanced by including more detailed information on the various grading and staging systems for liver biopsies, along with their historical context. This would provide additional depth and context to the review.
Despite this minor recommendation, I believe your review is an extremely useful and high-quality piece of material that is well worth publishing.

Author Response

Comment 1: I have had the opportunity to review manuscript, "Recent Advances in Histological and Molecular Evaluation of Liver Biopsies: An Updated Review," and would like to share my feedback. Overall, I believe this is an excellent work that deserves rapid publication.

Response 1: Thank you for your positive and encouraging feedback.

Comment 2: The work provides a deep and detailed analysis of the key aspects of liver biopsy evaluation. This is particularly valuable for trainees and specialists who are learning this subject. The text is written in a very clear and structured manner, making it accessible to a wide range of readers. The quality of the illustrations significantly enhances the reader's understanding and is a major strength of this manuscript.

Response 2: I appreciate your comments.

Comment 3: Recommendations for improvement: While the work is already very strong, I believe it could be further enhanced by including more detailed information on the various grading and staging systems for liver biopsies, along with their historical context. This would provide additional depth and context to the review.

Response 3: I greatly appreciate your kind remarks and thoughtful recommendation. In response, I have added comparison of several grading and staging systems and provided Table 6.

“Several histological scoring systems have been developed to evaluate the extent of necroinflammatory activity (grading) and fibrosis (staging) in chronic hepatitis [106-112]. Table 6 shows a comparison of these systems.”

Table 6. Comparison of scoring systems for grading and staging of chronic hepatitis.

System, Year of Publication

Grading (Necroinflammatory Activity)

Staging (Fibrosis)

Note

Reference

Knodell (HAI), 1981

Periportal/ bridging necrosis (score 0–10), intralobular degeneration and focal necrosis (score 0–4), portal inflammation (score 0–4)

Score 0–4 (no fibrosis to cirrhosis)

Introduced HAI; fibrosis included in total score (0–22); composite scoring can obscure the distinction between inflammation and fibrosis.

[72]

Scheuer, 1991

Portal/periportal necroinflammatory activity (grade 0–4), lobular necroinflammatory activity (grade 0–4)

Grade 0–4 (no fibrosis to cirrhosis)

Simplified semiquantitative assessment; independently scores necroinflammation and fibrosis

[106]

Desmet, 1994

Piecemeal necrosis and lobular activity (grade: minimal to severe)a)

Score 0–4 (no fibrosis to cirrhosis)

Chronic hepatitis is diagnosed based on etiology, grade, and stage

[107]

Ishak (modified HAI), 1995

Interface hepatitis (piecemeal necrosis) score 0–4), confluent necrosis (score 0–6), focal lytic necrosis/apoptosis/focal inflammation (score 0–4), portal inflammation (score 0–4); total score 0–18

Score 0–6 (no fibrosis to cirrhosis)

Improves Knodell HAI by separating grade/stage and adding detailed fibrosis scoring; standardized and reproducible histological assessment

[66]

Batts-Ludwig, 1995

Interface hepatitis (piecemeal necrosis) (grade 0–4), lobular inflammation and necrosis (grade 0–4)

Score 0–4 (no fibrosis to cirrhosis)

Provides a simplified, practical framework for grading and staging chronic hepatitis

[108]

METAVIR, 1996

Interface hepatitis (piecemeal necrosis) (score 0–3), lobular necrosis (score 0–2); overall histological activity (A0–A3) b)

Score F0–F4 (no fibrosis to cirrhosis)

Simplified scoring system; fibrosis evaluation is reliable

[73,74]

Park et al., 1999

Porto-periportal activity (score 0–4), lobular activity (score 0–4) c)

Score 0–4 (no fibrosis to cirrhosis)

Practical and reproducible; tailored for Korean clinical and pathological practice

[109]

Laennec, 2000

Not graded

Score F0–F4 (no fibrosis to cirrhosis); subdividing cirrhosis (F4) into F4A, F4B, or F4C d)

Correlated strongly with both the clinical stage of cirrhosis and the grade of portal hypertension

[110-112]

HAI, histological activity index; A, activity; F, fibrosis stage. a) Knodell HAI is used for grading (e.g., HAI 1–3 = minimal, HAI 4–8 = mild, HAI 9–12 = moderate, and HAI 13–18 = severe). b) METAVIR histological activity: A0 = no activity, A1 = mild activity, 2 = moderate activity, and A3 = severe activity. c) Porto-periportal activity is graded by assessing interface hepatitis (piecemeal necrosis) and portal inflammation, while lobular activity is graded based on lobular inflammation and necrosis. d) Cirrhosis (F4) is subdivided into F4A (mild), F4B (moderate), or F4C (severe) according to the thickness of fibrous septa and the size of regenerative nodules.

Comment 4: Despite this minor recommendation, I believe your review is an extremely useful and high-quality piece of material that is well worth publishing.

Response 4: I sincerely thank your kind endorsement and encouraging evaluation.